# LUCIDFLUX: CAPTION-FREE PHOTO-REALISTIC IMAGE RESTORATION VIA A LARGE-SCALE DIFFUSION TRANSFORMER

**Song Fei**[†]
The Hong Kong University of Science and Technology (Guangzhou)
sfei285@connect.hkust-gz.edu.cn

**Tian Ye**[†,‡]
The Hong Kong University of Science and Technology (Guangzhou)
tye610@connect.hkust-gz.edu.cn

**Lujia Wang**
The Hong Kong University of Science and Technology (Guangzhou)
eewanglj@hkust-gz.edu.cn

**Lei Zhu**[*]
The Hong Kong University of Science and Technology
The Hong Kong University of Science and Technology (Guangzhou)
leizhu@hkust-gz.edu.cn

(a) Image Restoration On Real-World Samples

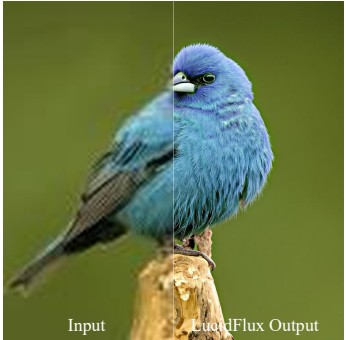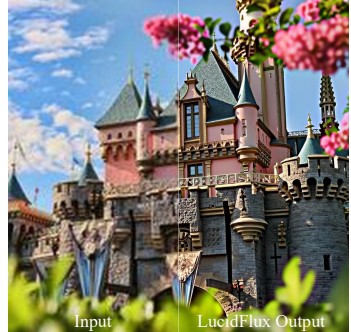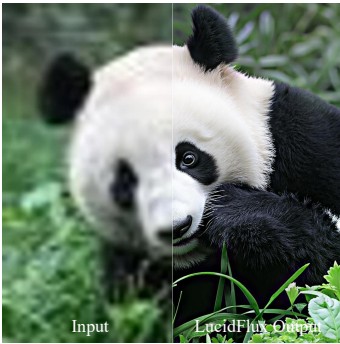

Input    LucidFlux Output    Input    LucidFlux Output    Input    LucidFlux Output

(b) Image Restoration On Different Scene

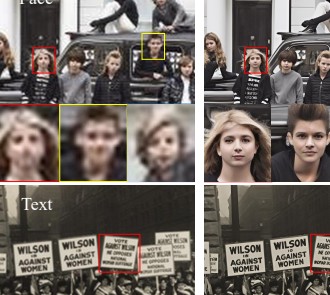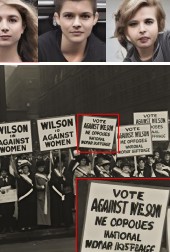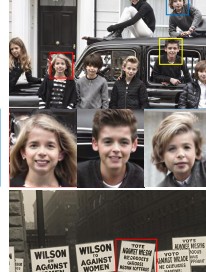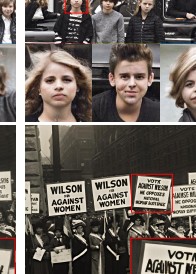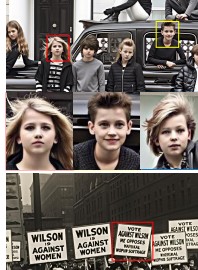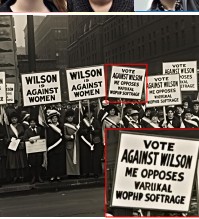

Face

Text

Low Quality Input    SeeSR    SUPIR    DreamClear    LucidFlux (Ours)

Figure 1: We present LucidFlux, a image restoration framework built on a large-scale diffusion transformer that delivers photorealistic restorations of real-world low-quality (LQ) images, outperforming state-of-the-art (SOTA) diffusion-based models across diverse degradations.

[†]Equal contribution    [‡]Project Leader    [*]Corresponding author.

## ABSTRACT

Image restoration (IR) aims to recover images degraded by unknown mixtures while preserving semantics—conditions under which discriminative restorers and UNet-based diffusion priors often oversmooth, hallucinate, or drift. We present **LucidFlux**, a caption-free IR framework that adapts a large diffusion transformer (Flux.1) without image captions. Our LucidFlux introduces a lightweight *dual-branch conditioner* that injects signals from the degraded input and a lightly restored proxy to respectively anchor geometry and suppress artifacts. Then, a *timestep- and layer-adaptive modulation* schedule is designed to route these cues across the backbone's hierarchy, in order to yield coarse-to-fine and context-aware updates that protect the global structure while recovering texture. After that, to avoid the latency and instability of text prompts or Vision-Language Model (VLM) captions, we enforce *caption-free semantic alignment* via SigLIP features extracted from the proxy. A scalable curation pipeline further filters large-scale data for structure-rich supervision. Across synthetic and in-the-wild benchmarks, our LucidFlux consistently outperforms strong open-source and commercial baselines, and ablation studies verify the necessity of each component. LucidFlux shows that, for large DiTs, *when, where, and what* to condition on—rather than adding parameters or relying on text prompts—is the governing lever for robust and caption-free image restoration in the wild.

**Project page:** `https://w2genai-lab.github.io/LucidFlux`
**Code:** `https://github.com/W2GenAI-Lab/LucidFlux`

# 1 INTRODUCTION

Images acquired in the wild exhibit mixed, unknown degradations—sensor noise, motion blur, lens aberrations, compression artifacts—that erode perceptual fidelity and induce semantic drift in recognition and analysis. *Image restoration (IR)* seeks to reconstruct images with high perceptual fidelity while preserving semantic consistency under such uncertainty and without access to degradation labels or side information. Despite steady progress, this combination of unknown mixtures, realism, and semantic preservation remains stubbornly challenging.

Discriminative restorers based on CNNs and Transformers (Ye et al. (2023; 2022); Chen et al. (2023c); Dong et al. (2016); Liang et al. (2021); Zamir et al. (2022); Chen et al. (2023b; 2022)) perform well on synthetic distortions but falter on in-the-wild mixtures, often oversmoothing textures or leaving visible artifacts. This gap has motivated generative approaches that leverage diffusion-based text-to-image priors to synthesize plausible structure and detail beyond the reach of purely discriminative models (Yu et al. (2024); Ai et al. (2024); Wu et al. (2024b); Wang et al. (2024a;b); Yue et al. (2023); Wu et al. (2024a); Lin et al. (2025a)). Early diffusion-based IR systems predominantly rely on Stable Diffusion (SD) UNet backbones (Rombach et al. (2022)), whose capacity and inductive bias tend to saturate under complex degradations, making it difficult to recover fine detail while maintaining global structure. More recent works have begun to explore transformer-based diffusion priors for restoration (e.g., DiT-based variants such as DreamClear Ai et al. (2024)), and we explicitly build on this line; in this paper we focus on how to adapt a large, generic DiT such as Flux.1 to real-world IR with lightweight, caption-free conditioning.

Recent advances in diffusion transformers (DiTs) open a promising avenue. In contrast to UNet architectures, DiTs employ attention-centric backbones that more effectively couple global context with local detail and carry richer generative priors. For instance, DreamClear (Ai et al. (2024)) builds on PixArt-$\alpha$ (Chen et al. (2023a)), a relatively small (0.6B) DiT, illustrating the promise of transformer backbones for restoration, and related DiT-based IR models are further reviewed in Sec. 2. However, their limited scale constrains robustness to mixed, real-world degradations and impedes the concurrent recovery of global structure and fine detail. Large-scale diffusion transformers such as Flux.1 (Labs (2024)) deliver strong modeling capacity for diverse, mixed-degradation image restoration, yet ***direct transfer rarely works off-the-shelf***. Previous ControlNet-style conditioning methods (Yu et al. (2024); Ai et al. (2024); Zhang et al. (2023)) ***disrupt the parameter–structure balance*** and underutilize the backbone's temporal and hierarchical division of labor. Unconstrained

injection of degraded observations amplifies artifacts; relying on VLM-generated captions further *increases latency and risks semantic drift*[0]. Meanwhile, backbones at this scale are ***decisively data-limited***: gains follow data–compute scaling only when trained on ***curated, large-scale, high-quality*** sets. Public web corpora fall short for IR—they skew toward aesthetic, compression-heavy images, contain substantial near-duplicates and low-information frames, and rarely cover the long-tail mixtures of real degradations or provide usable pairs. Without rigorous filtering and structure-aware selection, large DiTs underutilize capacity and overfit spurious artifacts, underscoring the need for an explicit curation pipeline. Taken together, these tensions point to a more structured path, one that schedules conditioning across timesteps and layers, couples robust input handling with caption-free inference, and remains practical to assemble on available datasets.

To operationalize this path, we introduce ***LucidFlux***, a caption-free IR framework that adapts the large-scale Flux.1 diffusion transformer to restoration. The core of our ***LucidFlux*** is a *lightweight dual-branch conditioner*—a two-block transformer module that injects signals from the degraded input without inflating the parameters. One branch ingests the low-quality image to anchor the geometry and layout, while the other consumes a lightly restored proxy to suppress hard artifacts; their outputs are scheduled through a *timestep- and layer-adaptive modulation* that aligns guidance with the backbone's hierarchical roles, yielding coarse-to-fine, context-aware updates that preserve texture while protecting global structure. To avoid the latency and drift introduced by text prompts, we enforce semantic consistency via *caption-free alignment with SigLIP*, extracting semantic cues directly from the proxy. We pair the model with an automated three-stage curation pipeline—blur detection, flat-region filtering, and perceptual quality scoring—to assemble diverse training sets at the *billion-parameter scale.*

Our contributions are as follows:

- **LucidFlux framework**. We adapt a large diffusion transformer (Flux.1) to IR with a lightweight dual-branch conditioner and timestep- and layer-adaptive modulation, aligning conditioning with the backbone's hierarchical roles while keeping less trainable parameters.

- **Caption-free semantic alignment**. A SigLIP-based module preserves semantic consistency without prompts or captions, mitigating latency and semantic drift.

- **Scalable data curation pipeline.** A reproducible, three-stage filtering pipeline yields diverse, structure-rich datasets that scale to billion-parameter training.

- **State-of-the-art results**. LucidFlux sets new SOTA on a broad suite of benchmarks and metrics, surpassing competitive open- and closed-source baselines; ablation studies confirm the necessity of each module.

## 2 RELATED WORK

**Generative Priors for IR.** Large-scale pretrained generative models, particularly text-to-image diffusion models (Rombach et al. (2022); Podell et al. (2023); Yu et al. (2025); Yang et al. (2025)), have shown strong capability in synthesizing high-fidelity textures and structures for image restoration. Existing approaches build on different backbones, with SUPIR (Yu et al. (2024)) using SDXL, DreamClear (Ai et al. (2024)) relying on PixArt-$\alpha$ (Chen et al. (2023a)), StableSR on SD, SeeSR on SD2, and Resshift (Yue et al. (2023)) and SinSR (Wang et al. (2024b)) trained from scratch. UNet-based systems (e.g., StableSR, SeeSR, Resshift, SinSR, SDXL-based variants) inherit the capacity and inductive-bias limitations of SD-style backbones under complex, mixed degradations, while recent DiT-based IR models such as DreamClear demonstrate the promise of transformer priors but typically adopt relatively small, task-specific DiTs and heavy ControlNet-style adapters. This makes it difficult to fully exploit the capacity of modern large-scale text-to-image transformers for IR. Addressing these challenges, we propose **LucidFlux**, an image restoration framework that adapts the 12B Flux.1 DiT with a lightweight dual-branch conditioner and caption-free conditioning, providing a different operating point within this line of generative-prior-based IR methods.

---

[0]Appendix Sec. A.1 quantifies the prevalence of degradation-related terms in VLM captions, and Appendix Fig. 5 demonstrates how such bias can misguide restoration.

**Semantic Alignment.** Preserving semantic fidelity during image restoration is a significant challenge. Existing methods often rely on generating captions from degraded images via vision–language models at inference time (Yu et al. (2024); Ai et al. (2024); Kong et al. (2025)), which causes additional computational cost and may produce inconsistencies between training and inference. Moreover, such captions can explicitly encode degradation-related terms, introducing semantic bias from artifacts rather than underlying content (see Appendix Sec. A.1). Alternative strategies employ coarse textual cues (Wu et al. (2024b;a)), but such signals are generally insufficient to capture fine-grained semantic content. In contrast, LucidFlux leverages a caption-free SigLIP-based semantic alignment module that extracts features from lightly restored proxies and projects them through a lightweight Connector into the text-conditioning space expected by Flux.1, facilitating caption-free guidance without additional VLM calls and avoiding degradation-related caption bias, and ensuring that restored outputs maintain high semantic consistency without hallucinations.

## 3 METHODOLOGY

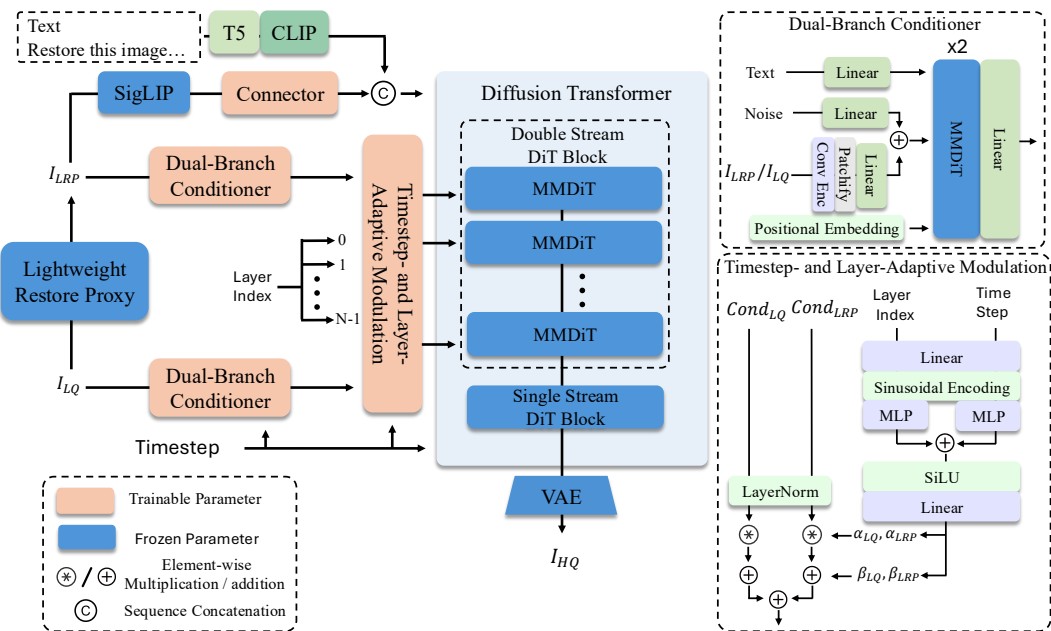

Figure 2: Overview of the proposed architecture for image restoration. Our method integrates dual condition streams (LQ and LRP) with timestep- and layer-adaptive modulation modules, and incorporates SigLIP semantic priors through a connector into a Flux-based DiT backbone to jointly enhance perceptual quality and semantic consistency.

Our framework is built upon a Flux-based DiT backbone, augmented with two parallel ControlNet branches. The first branch processes the original low-quality image (LQ), while the second branch takes a lightly restored version of the input (LRP) generated by a lightweight restoration model. Both streams capture complementary information, which is subsequently modulated through timestep- and layer-adaptive modules to align with the DiT feature space. Moreover, we incorporate semantic priors extracted from SigLIP and enhanced with a connector, which are injected into the DiT layers to facilitate semantic consistency and fine-grained texture restoration.

Practically, we eschew inference-time captions: Appendix Sec. A.1 quantifies that 17–24% of VLM captions introduce degradation-related terms, and Appendix Fig. 5 shows such bias misguides restoration, degrading perceptual quality.

### 3.1 LIGHTWEIGHT DUAL-BRANCH CONDITIONER

Directly conditioning on the low-quality (LQ) image preserves high-frequency details but often leaks residual artifacts under mixed degradations; conditioning on a lightly restored proxy (LRP)

suppresses artifacts but tends to oversmooth textures. Following the dual-branch paradigm proposed in (Ai et al. (2024)), we therefore decouple *structure anchoring* and *artifact suppression* into two signals and encode them with a *minimal-overhead* conditioner that interfaces with the Flux.1 DiT backbone without duplicating large blocks.

As illustrated in the top-right of Fig. 2, only the core conditioning pathway is shown for clarity, while other components such as timestep embeddings remain consistent with Flux. Throughout, the Flux.1 backbone and VAE remain frozen for stability and efficiency. The output feature map $I_{\text{LRP}}$ of the LRP is computed by:

$$I_{\text{LRP}} = \text{LRP}(I_{\text{LQ}}). \tag{1}$$

Both $I_{\text{LQ}}$ and $I_{\text{LRP}}$ are processed by the *Dual-Branch Conditioner (DBC)*, which then converts each input into compact conditioning tokens through a two-block MMDiT applied at latent resolution,

$$\phi_{\text{LQ}} = \text{DBC}(I_{\text{LQ}}), \qquad \phi_{\text{LRP}} = \text{DBC}(I_{\text{LRP}}), \tag{2}$$

where a simple 8-layer 3x3 convolutional encoder (Conv Enc) maps the input image to the VAE latent space, then *patchifies* the latent and projects patches into a 2D-positioned sequence before two stacked transformer blocks; *weights are not shared across branches*, as the LQ stream emphasizes detail-preserving, noise-tolerant cues while the LRP stream favors structure-first, artifact-suppressed representations. Using separate parameters avoids competing gradients and preserves branch complementarity, while the conditioner remains minimal (two blocks per branch, constant overhead w.r.t. layer depth and far smaller than ControlNet-style duplication of a large DiT). Intuitively, $\phi_{\text{LQ}}$ carries detail-preserving yet noisy cues, whereas $\phi_{\text{LRP}}$ provides artifact-robust structure; the subsequent *timestep- and layer-adaptive condition modulation* (Sec. 3.2) consumes these two complementary signals for coarse-to-fine, context-aware guidance without increasing the conditioner's footprint.

## 3.2 Timestep- and Layer-Adaptive Condition Modulation

Diffusion transformers exhibit a temporal–hierarchical division of labor: early timesteps reconstruct coarse structures while later ones refine high-frequency details; similarly, shallower layers capture low-level edges and deeper layers process semantics (Park et al. (2023); Qian et al. (2024)). Applying identical conditioning across all timesteps and layers risks redundancy or conflict. We therefore modulate the outputs of the *dual-branch conditioner (DBC)* in a way that is *adaptive to both timestep $t$ and layer index $l$* while keeping the heavy Flux.1 backbone frozen. A lightweight modulation head takes sinusoidally encoded $(t/T, l/L)$ and predicts *feature-wise* (per-channel) scale and bias for each branch independently:

$$\alpha_{\text{m}}^{t,l}, \beta_{\text{m}}^{t,l} = \text{Modulation}\big(\text{PE}(t/T, l/L)\big), \quad \text{m} \in \{\text{LQ}, \text{LRP}\}, \ \alpha_{\text{m}}^{t,l}, \beta_{\text{m}}^{t,l} \in \mathbb{R}^{d_c}. \tag{3}$$

These parameters affect an AdaptiveLN-style adjustment,

$$\tilde{\phi}_{\text{LQ}}^{t,l} = \alpha_{\text{LQ}}^{t,l} \odot \phi_{\text{LQ}} + \beta_{\text{LQ}}^{t,l}, \qquad \tilde{\phi}_{\text{LRP}}^{t,l} = \alpha_{\text{LRP}}^{t,l} \odot \phi_{\text{LRP}} + \beta_{\text{LRP}}^{t,l}, \tag{4}$$

We then fuse the branches without additional normalization:

$$\text{Cond}^{t,l} = \phi_{\text{LQ}}^{t,l} + \phi_{\text{LRP}}^{t,l}. \tag{5}$$

Predicting $\alpha/\beta$ *per channel* supplies sufficient flexibility to track the backbone's roles across $t$ and $l$ without inflating capacity, and *independent* modulation for LQ vs. LRP preserves their complementary inductive biases (detail-preserving vs. artifact-robust). Keeping modulation inside the lightweight conditioner maintains negligible overhead while enabling coarse-to-fine, timestep- and layer-aware guidance; ablations (Sec. 4.3) show that removing either the temporal or the hierarchical dependency degrades fidelity under mixed degradations.

## 3.3 SigLIP for Caption-Free Semantic Alignment

Text-to-image (T2I) diffusion models are typically conditioned on text, and many restoration methods adopt captions as semantic guidance (Ai et al. (2024); Yu et al. (2024)). During training, such captions are often derived from clean ground truth, yielding idealized supervision. At inference, however, only degraded inputs are available; captions generated from low-quality images tend to inherit degradation-specific artifacts and, when produced by large VLMs, add substantial latency

and exacerbate a train–test mismatch (Sun et al. (2024)). We replace caption generation with a *caption-free* semantic pathway. Concretely, we extract image semantics from the lightly restored proxy $I_{\text{LRP}}$ using a frozen SigLIP encoder and map them into the backbone's textual embedding space via a lightweight *Connector*:

$$z_s = \text{Connector}\big(\text{SigLIP}(I_{\text{LRP}})\big). \tag{6}$$

The projected semantics $z_s$ are concatenated with a small set of prompt tokens $c$ (default instruction) to form the multimodal context fed to the DiT backbone:

$$Context = \text{Concat}(z_s, c). \tag{7}$$

Grounding semantics in $I_{\text{LRP}}$ stabilizes content under mixed degradations, while the Connector furnishes a drop-in bridge to the text-conditioning interface, avoiding any duplication of heavy modules. This design eliminates external captions at both training and inference, reducing latency and removing a major source of caption-induced semantic variance (e.g., differences across captioners or paraphrases). Grounding semantics in $z_s$ keeps outputs structurally faithful and semantically aligned to the input.

## 3.4 SCALING UP REAL-WORLD HIGH-QUALITY DATA FOR IMAGE RESTORATION

Although large-scale T2I diffusion models are pretrained on hundreds of millions of image–text pairs, they are not tailored for the image restoration task of our work. Training large diffusion transformers for IR requires *task-aligned data at scale with strong structure and perceptual quality*. However, publicly available restoration corpora remain modest and/or lack reproducible quality control: DIV2K (800/100) (Agustsson & Timofte (2017)), Flickr2K (2,650) (Agustsson & Timofte (2017)), LSDIR ($\approx$ 85K with manual curation) (Li et al. (2023)), and SUPIR (20M without disclosed filtering criteria) (Yu et al. (2024)), while DreamClear (Ai et al. (2024)) synthesizes 1M pairs at substantial computational cost. This leaves a practical gap between what large DiTs need and what current datasets provide.

To bridge this gap, we introduce, to our knowledge, the first publicly documented and extensively validated *IR-specific data filtering pipeline*. It is *fully automatic* (parameters empirically set, pipeline automatic once fixed) and comprises three stages—blur screening, flat-region suppression, and perceptual-quality ranking—explicitly designed to retain structure-rich, high-quality images while discarding unsuitable samples.

**Data source.** Our initial dataset is collected from two sources. First, we collect 2.3M images from the Internet, specifically from Pexels[1] and Unsplash[2], using their official image APIs. We query both platforms with 23,914 unique keywords extracted from the Unsplash Lite Dataset[3], retrieve the associated metadata, and download images via the official URLs provided. This keyword-based pipeline ensures fully reproducible data collection without bypassing platform restrictions. The full keyword list is included in the supplementary materials as `keywords.txt`. In addition, we incorporate 557K images from the Photo-Concept-Bucket dataset (bghira (2023)), yielding a total of 2.9M candidate images. This combined pool serves as the raw data for subsequent filtering.

**Blur detection.** Images that are heavily blurred or contain excessive high-frequency noise provide unreliable structural cues and are thus unsuitable for training. Following LSDIR (Li et al. (2023)), we quantify the degree of blur using the variance of the Laplacian $S_{\text{blur}}(I) = \text{Var}\big(\nabla^2 I\big)$, where $I$ denotes an input image. Only images with $150 \leq S_{\text{blur}}(I) \leq 8000$ (Li et al. (2023)) are retained, effectively excluding both overly blurred and noisy samples. These bounds are empirically hand-tuned based on preliminary experiments and careful visual audits on a held-out subset to balance removal of extreme blur/noise while retaining legitimate shallow-depth-of-field and low-light scenes.

**Flat-region detection.** Images dominated by textureless regions may bias the model towards producing over-smoothed outputs. To mitigate this, each image is divided into non-overlapping $240 \times 240$ patches, and the edge richness of each patch is measured using the Sobel operator with $S_{\text{flat}} = \text{Var}\Big(\sqrt{(\partial_x I)^2 + (\partial_y I)^2}\Big)$. Patches with $S_{\text{flat}} < 800$ are considered textureless, and images containing more than 50% such patches are discarded. Both the 800 patch-level threshold and

---

[1] `https://www.pexels.com/`
[2] `https://unsplash.com/`
[3] `https://github.com/unsplash/datasets.git`

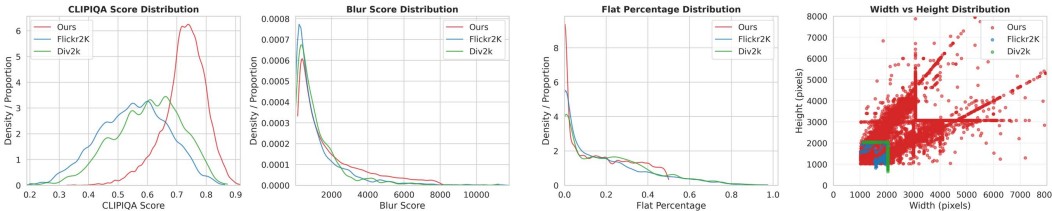

Figure 3: Comparison of dataset attributes. Our dataset exhibits higher CLIP-IQA scores, lower flatness, and more diverse resolutions than Flickr2K (Agustsson & Timofte (2017)) and DIV2K (Agustsson & Timofte (2017)).

the 50% image-level ratio are empirically set by manual inspection of edge-statistics distributions and visual audits; they provide a conservative balance that suppresses large flat backgrounds yet preserves natural sky/water regions. This ensures that retained images exhibit sufficient edge and texture diversity, essential for high-fidelity restoration. After applying blur and flat-region filtering, 1.28M candidate images remain for the final image quality assessment (IQA) filtering stage.

**IQA Filtering for High-quality Data.** While LSDIR employs manual curation in its final stage, such human intervention is impractical for scaling to larger datasets. We apply CLIP-IQA to further ensure perceptual quality of our training data. The remaining images are ranked by their perceptual scores $s_i$, and only the top 20% are retained, i.e., $\{i \mid s_i \geq \text{quantile}_{0.8}(\{s_i\})\}$, resulting in 257K high-quality images. The 20% cutoff is empirically chosen after careful inspection at multiple percentiles (e.g., 10/20/30%), trading off perceptual quality against semantic/content diversity. By additionally incorporating 84K high-quality samples from LSDIR (Li et al. (2023)), the final curated dataset comprises 342K high-quality images. Once these cutoffs are fixed, the pipeline executes fully automatically at scale. For generating paired training data, degraded counterparts are synthesized using the Real-ESRGAN degradation pipeline (Wang et al. (2021)) as implemented in (Ai et al. (2024)), across 4 epochs, producing a total of 1.36M image pairs. This procedure ensures both diversity and realism in the low-quality inputs, facilitating effective model training. To assess the effectiveness of our filtered data, we randomly select 10K samples and compare their attribute distributions with existing datasets. Figure 3 shows that our dataset achieves higher CLIP-IQA scores, comparable blur scores, lower flatness values that reflect richer textures, and more diverse resolutions than Flickr2K and DIV2K. In Appendix 6, we also analyze semantic diversity using t-SNE, and it shows that our dataset demonstrates substantially broader semantic coverage.

## 4 EXPERIMENT

### 4.1 IMPLEMENTATION DETAILS

We train a large Flux-based generative model, LucidFlux, while *freezing all blocks of the Flux backbone* and training only the task-specific modules introduced by our method. Freezing the backbone stabilizes optimization and prevents catastrophic forgetting, while concentrating capacity on the new modules that realize our objective. Under a standard $L_2$ latent loss for velocity prediction, as commonly used in flow-matching models (Labs et al. (2025)), training runs on 8×NVIDIA A800 GPUs with DeepSpeed ZeRO-2. We choose ZeRO-2 because it shards optimizer states and gradients—dramatically reducing memory footprint—without partitioning model parameters, which preserves simple forward passes and yields higher throughput than ZeRO-3 in our setting. This enables larger activation budgets at 1024×1024 resolution and steady scaling with modest communication overhead. We use Adafactor (Shazeer & Stern (2018)) with a learning rate of $2\times10^{-5}$ and weight decay 0.01. The per-GPU batch size is 2 with gradient accumulation of 2 steps, giving an effective batch size of 32 across 8 GPUs. We resume our Siglip connector based on Flex.1-alpha-Redux checkpoint. The full training completes in approximately 7 GPU-days. Following many existing works, we employ SwinIR (Liang et al. (2021)) as a lightweight restore proxy.

Table 1: Quantitative comparison across different IQA metrics on RealSR (Wu et al. (2024b)), RealLQ250 (Ai et al. (2024)), DIV2K-Val, LSDIR-Val and DRealSR.

| Benchmark | | Metric | ResShift | StableSR | SinSR | DiffBIR | SeeSR | DreamClear | SUPIR | LucidFlux(Ours) |
|---|---|---|---|---|---|---|---|---|---|---|
| **Caption-Free** | | | ✓ | ✓ | ✓ | ✓ | ✗ | ✗ | ✗ | ✓ |
| **Real-world** | **DRealSR** | CLIP-IQA+ ↑ | 0.4655 | 0.3732 | 0.5402 | 0.6475 | 0.6257 | 0.4461 | 0.5494 | **0.6748** |
| | | Q-Align ↑ | 2.6311 | 2.1245 | 3.1334 | 3.0490 | 3.2745 | 2.4213 | 3.4720 | **3.6919** |
| | | MUSIQ ↑ | 40.9795 | 29.6691 | 53.9138 | 60.0759 | 61.3222 | 35.1911 | 54.9279 | **66.6833** |
| | | MANIQA ↑ | 0.2687 | 0.2402 | 0.3455 | 0.4900 | 0.4505 | 0.2675 | 0.3482 | **0.4985** |
| | | NIMA ↑ | 4.3178 | 3.9048 | 4.6226 | 4.6543 | 4.6401 | 3.9368 | 4.5063 | **4.9625** |
| | | CLIP-IQA ↑ | 0.4964 | 0.3383 | 0.6631 | 0.6781 | 0.6760 | 0.4360 | 0.5309 | **0.6879** |
| | | NIQE ↓ | 10.3005 | 8.6022 | 6.9800 | 6.4852 | 6.4502 | 7.0163 | 5.9091 | **4.7034** |
| | **RealSR** | CLIP-IQA+ ↑ | 0.5005 | 0.4408 | 0.5416 | 0.6543 | 0.6731 | 0.5331 | 0.5640 | **0.7074** |
| | | Q-Align ↑ | 3.1045 | 2.5087 | 3.3615 | 3.3157 | 3.6073 | 3.0044 | 3.4682 | **3.7555** |
| | | MUSIQ ↑ | 49.50 | 39.98 | 57.95 | 61.7750 | 67.57 | 49.48 | 55.68 | **70.20** |
| | | MANIQA ↑ | 0.2976 | 0.2356 | 0.3753 | 0.4744 | 0.5087 | 0.3092 | 0.3426 | **0.5437** |
| | | NIMA ↑ | 4.7026 | 4.3639 | 4.8282 | 4.8192 | 4.8957 | 4.4948 | 4.6401 | **5.1072** |
| | | CLIP-IQA ↑ | 0.5283 | 0.3521 | 0.6601 | 0.6805 | **0.6993** | 0.5390 | 0.4857 | 0.6783 |
| | | NIQE ↓ | 9.0674 | 6.8733 | 6.4682 | 6.0700 | 5.4594 | 5.2873 | 5.2819 | **4.2893** |
| | **RealLQ250** | CLIP-IQA+ ↑ | 0.5529 | 0.5804 | 0.6054 | 0.6918 | 0.7034 | 0.6810 | 0.6532 | **0.7406** |
| | | Q-Align ↑ | 3.6318 | 3.5586 | 3.7451 | 3.9757 | 4.1423 | 4.0640 | 4.1347 | **4.3935** |
| | | MUSIQ ↑ | 59.50 | 57.25 | 65.45 | 67.5313 | 70.38 | 67.08 | 65.81 | **73.01** |
| | | MANIQA ↑ | 0.3397 | 0.2937 | 0.4230 | 0.4899 | 0.4895 | 0.4400 | 0.3826 | **0.5589** |
| | | NIMA ↑ | 5.0624 | 5.0538 | 5.2397 | 5.3132 | 5.3146 | 5.2200 | 5.0806 | **5.4836** |
| | | CLIP-IQA ↑ | 0.6129 | 0.5160 | **0.7166** | 0.7137 | 0.7063 | 0.6950 | 0.5767 | 0.7122 |
| | | NIQE ↓ | 6.6326 | 4.6236 | 5.4425 | 5.1193 | 4.4383 | 3.8700 | **3.6591** | 3.6742 |
| **Synthetic** | **DIV2K-Val** | CLIP-IQA+ ↑ | 0.5583 | 0.5760 | 0.6128 | 0.6973 | 0.7116 | 0.6585 | 0.6719 | **0.7492** |
| | | Q-Align ↑ | 3.5761 | 3.4226 | 3.7336 | 3.8509 | 3.9323 | 3.9323 | 4.1659 | **4.5311** |
| | | MUSIQ ↑ | 60.5932 | 57.4246 | 66.0906 | 69.1822 | 71.4947 | 65.8187 | 67.9074 | **73.9045** |
| | | MANIQA ↑ | 0.3421 | 0.2902 | 0.4341 | 0.5015 | 0.5104 | 0.4369 | 0.4148 | **0.5819** |
| | | NIMA ↑ | 5.0430 | 5.0341 | 5.1810 | 5.1941 | 5.2709 | 5.1663 | 5.1516 | **5.4884** |
| | | CLIP-IQA ↑ | 0.6017 | 0.5002 | **0.7166** | 0.7143 | 0.7149 | 0.6663 | 0.5848 | 0.7034 |
| | | NIQE ↓ | 6.1976 | 4.9810 | 5.3679 | 4.8437 | 4.2823 | 4.1634 | 3.7701 | **3.7283** |
| | | PSNR ↑ | 18.3802 | 18.3269 | 18.0956 | **20.0389** | 18.2529 | 17.5701 | 17.7567 | 15.4393 |
| | | SSIM ↑ | 0.4394 | **0.4819** | 0.4259 | 0.5242 | 0.4684 | 0.4291 | 0.4482 | 0.3837 |
| | | LPIPS ↓ | 0.3738 | 0.3933 | 0.3919 | 0.3582 | **0.3497** | 0.3621 | 0.3785 | 0.4312 |
| | **LSDIR-Val** | CLIP-IQA+ ↑ | 0.5248 | 0.5576 | 0.5582 | 0.6977 | 0.7258 | 0.6995 | 0.7126 | **0.7440** |
| | | Q-Align ↑ | 3.5317 | 3.4878 | 3.7095 | 3.9514 | 4.2997 | 4.2391 | 4.3468 | **4.5959** |
| | | MUSIQ ↑ | 57.6691 | 57.0838 | 63.9586 | 68.4680 | 72.0142 | 70.7186 | 70.3340 | **74.1923** |
| | | MANIQA ↑ | 0.3408 | 0.2990 | 0.4131 | 0.5356 | 0.5529 | 0.5059 | 0.4482 | **0.5979** |
| | | NIMA ↑ | 5.0916 | 5.0628 | 5.3353 | 5.3566 | 5.4245 | 5.3773 | 5.3692 | **5.6221** |
| | | CLIP-IQA ↑ | 0.5691 | 0.4991 | **0.6766** | 0.7066 | 0.7314 | 0.6941 | 0.6105 | 0.6836 |
| | | NIQE ↓ | 6.4447 | 4.2104 | 5.1771 | 4.4015 | 3.9402 | 3.3318 | **2.9610** | 3.5571 |
| | | PSNR ↑ | **17.3040** | 17.1480 | 16.8241 | 18.8862 | 17.0782 | 16.2114 | 16.1598 | 14.8688 |
| | | SSIM ↑ | 0.3935 | 0.4026 | 0.3710 | 0.4652 | **0.4113** | 0.3823 | 0.3636 | 0.3697 |
| | | LPIPS ↓ | 0.4824 | 0.4655 | 0.4637 | 0.4344 | 0.3969 | **0.3720** | 0.4408 | 0.4148 |

## 4.2 COMPARISON WITH STATE-OF-THE-ART METHODS

We evaluate our approach against several state-of-the-art diffusion-based methods, including ResShift (Yue et al. (2023)), StableSR (Wang et al. (2024a)), SinSR (Wang et al. (2024b)), Diff-BIR (Lin et al. (2024)), SeeSR (Wu et al. (2024b)), SUPIR (Yu et al. (2024)), and DreamClear (Ai et al. (2024)). Following many existing works (Wang et al. (2024a); Wu et al. (2024b); Ai et al. (2024); Yu et al. (2024)), experiments are conducted on both synthetic and real-world benchmark datasets. For the synthetic data, we randomly crop 2,124 patches from the validation sets of DIV2K (Agustsson & Timofte (2017)) and LSDIR (Li et al. (2023)). For DIV2K, we use the five original degradation types: bicubic, unknown, mild, difficult, and wild. LSDIR-Val is generated by applying the same degradation pipeline used during training. For the real-world data, we adopt center-cropped images from RealSR (Cai et al. (2019)), DRealSR (Wei et al. (2020)) as used in (Wu et al. (2024b)) and RealLQ250 (Ai et al. (2024)). All evaluations are performed at a resolution of $1024 \times 1024$, with all super-resolution methods using an upscale factor of 4.

**Metrics.** We evaluate all methods using seven no-reference image quality assessment metrics, including CLIP-IQA+ (Wang et al. (2023)), Q-Align (Wu et al. (2023)), MUSIQ (Ke et al. (2021)), MANIQA (Yang et al. (2022)), NIMA (Talebi & Milanfar (2018)), CLIP-IQA (Wang et al. (2023)), and NIQE (Zhang et al. (2015)), as well as three reference-based metrics including PSNR, SSIM (Wang et al. (2004)), and LPIPS (Zhang et al. (2018)). These metrics assess the restoration performance across perceptual quality, semantic alignment, and structural fidelity.

**Qualitative Comparisons.** Figure 4 presents visual comparisons on representative samples from RealLQ250. SeeSR and DreamClear reduce some degradations but tend to leave residual artifacts or produce oversmoothed outputs with limited texture recovery. SUPIR generates cleaner results yet often loses fine details, leading to overly smooth surfaces. In contrast, our method achieves clearer edges, richer textures, and better semantic consistency with the degraded inputs, especially in challenging regions such as hair, text, and high-frequency patterns. These qualitative observations align with the quantitative results in Table 1, further confirming the effectiveness of our approach.

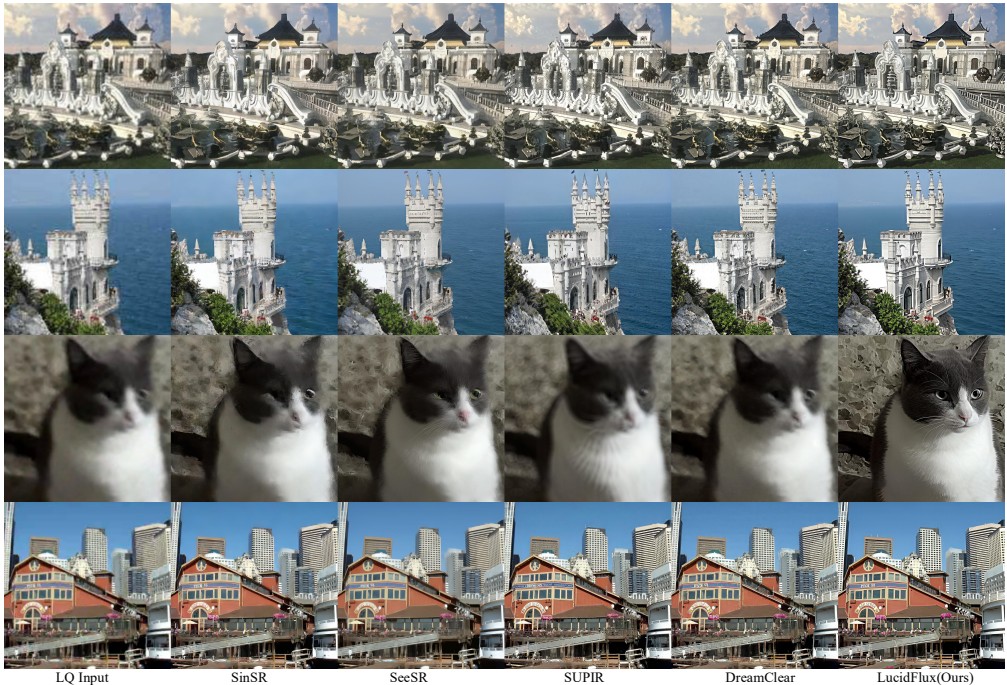

<table>
<tr><td>LQ Input</td><td>SinSR</td><td>SeeSR</td><td>SUPIR</td><td>DreamClear</td><td>LucidFlux(Ours)</td></tr>
</table>

Figure 4: Qualitative comparisons on RealLQ250. Baseline methods either leave noticeable artifacts or yield over-smoothed textures, while our approach restores sharper details. See Figure 9 to Figure 14 in Appendix for more visual comparisons.

**Quantitative Comparisons.** Table 1 reports the IQA metric results on real-world and synthetic benchmarks. Our method consistently outperforms prior approaches on perceptual and semantic-oriented metrics, such as CLIP-IQA+, MUSIQ, MANIQA, Q-Align, and NIMA, highlighting its ability to generate visually faithful and semantically aligned restorations. On real-world datasets (e.g., DRealSR, RealSR, RealLQ250), LucidFlux achieves clear gains over existing caption- or tag-based methods. For distortion-focused measures like PSNR and SSIM on synthetic datasets, prior approaches report slightly higher values, yet these metrics are widely recognized as being less correlated with human perceptual quality. In contrast, our method delivers state-of-the-art performance on modern IQA benchmarks, supporting the view that advanced IR frameworks should be evaluated with perceptual and semantic quality measures rather than traditional distortion metrics.

**Runtime and Model Scale Comparison.** We compare LucidFlux with SeeSR, SUPIR, and Dream-Clear in terms of runtime and model size in Table 3. Despite using a substantially larger backbone (12B), our LucidFlux achieves a competitive total runtime by eliminating the caption preprocessing. In contrast, SeeSR, SUPIR, and DreamClear require additional preprocessing and rely on smaller backbones (1.29B, 3.5B, 0.6B), resulting in higher latency relative to their size. For trainable adapters, LucidFlux maintains a balanced design (1.6B), outperforming SUPIR (1.3B) in representational capacity while remaining more efficient than DreamClear (2.2B).

**Comparison with Close-Source Commercial Methods.** We further compare our LucidFlux with several widely used commercial image restoration solutions, including HYPIR-FLUX (Group (2025)), Seedream 4.0 (ByteDance Seed Vision Team (2025)), Topaz (Labs (2025)), Gemini-NanoBanana (DeepMind (2025)), and MeiTu SR (MeiTu (2025)). For clarity, these baselines consist of (i) specialized SR/restoration products (HYPIR-FLUX, Topaz, MeiTu SR) that users rely on in practice, and (ii) unified generative or multimodal systems (Seedream 4.0, Gemini-NanoBanana) that also expose restoration/upscaling interfaces and can handle the same degraded inputs. We include the latter not as dedicated IR baselines, but to contextualize LucidFlux against the current generation of widely deployed unified commercial models. All evaluations are conducted under the same experimental settings, and the identical IQA metrics are used as in the open-source comparisons. Table 2 reports the quantitative results of different methods. Our LucidFlux achieves the

Table 2: Quantitative comparison across different IQA metrics with commercial models on RealLQ250.

| Method | CLIP-IQA+ ↑ | Q-Align ↑ | MUSIQ ↑ | MANIQA ↑ | NIMA ↑ | CLIP-IQA ↑ | NIQE ↓ |
|---|---|---|---|---|---|---|---|
| LQ Input | 0.6218 | 2.1693 | 44.1541 | 0.3718 | 3.8664 | 0.6079 | 6.0790 |
| Seedream 4.0 | 0.5002 | 3.6931 | 52.3771 | 0.2794 | 4.7024 | 0.4124 | 4.9393 |
| Gemini-NanoBanana | 0.3780 | 3.3114 | 44.6310 | 0.2548 | 4.6571 | 0.4434 | 6.0865 |
| MeiTu SR | 0.6653 | 4.1464 | 66.5936 | 0.4498 | 5.2103 | 0.6663 | 5.4125 |
| LucidFlux (Ours) | **0.7406** | **4.3935** | **73.01** | **0.5589** | **5.4836** | **0.7122** | **3.6742** |

Table 3: Runtime (s) and parameter scale (B).

| | SeeSR | SUPIR | DreamClear | LucidFlux |
|---|---|---|---|---|
| Caption (s) | 0.100 | 5.900 | 8.700 | 0.012 |
| Inference (s) | 22.380 | 16.600 | 28.900 | 23.612 |
| Total (s) | 22.480 | 22.500 | 37.600 | 23.612 |
| Backbone (B) | 1.29 | 2.6 | 0.6 | 12 |
| Adapter (B, train.) | 1.6 | 1.3 | 2.2 | 1.6 |
| Total (B) | 2.89 | 3.9 | 2.8 | 13.6 |

Table 4: Ablation study on RealLQ250. Evaluation metrics for three main contributions of our method.

| Setting | CLIP-IQA | CLIP-IQA+ | MUSIQ |
|---|---|---|---|
| Dual-Branch Conditioner Only | 0.585 | 0.609 | 61.582 |
| + SigLIP Alignment | 0.600 | 0.620 | 62.000 |
| + TLCM | 0.622 | 0.635 | 65.500 |
| + Large HQ Data (Our method) | **0.7122** | **0.7406** | **73.0088** |

largest scores across all metrics and outperforms other commercial solutions. MeiTu SR shows the best performance among compared methods, but its restoration results generally have less details than our LucidFlux. In contrast, our method balances strong quantitative performance with reliable and consistent restoration, which makes it particularly suitable for real-world applications. See our Appendix Figure 8 for qualitative comparisons.

## 4.3 ABLATION STUDY

We ablate our three contributions in testing RealLQ250 and report the quantitative results in Table 4. Starting from the Dual-Branch Conditioner (DBC) trained on LSDIR, our CLIP-IQA / CLIP-IQA+ / MUSIQ scores are 0.585/0.609/61.582, and all three scores are improved after adding caption-free SigLIP semantic alignment. Our timestep- and layer-adaptive condition modulation (TLCM) further improves score performance, and scaling to our curated large-scale high-quality data provides the largest jump over TLCM on three metrics. The progression indicates that SigLIP alignment stabilizes semantics; TLCM exploits the DiT hierarchy; and data curation supplies structure-rich supervision, and thus all three modifications on DBC are required for the final outcome.

**Caption-Free vs. Caption-Based Semantic Pathways.** To isolate the effect of the caption-free design, we evaluate three variants built on the same Flux.1 backbone: (i) GT captions (LLaVA on the ground-truth clean images), (ii) VLM captions (LLaVA on the lightly restored proxy), and (iii) caption-free (ours) using the SigLIP semantic pathway without captions. Full quantitative results and qualitative comparisons are provided in Appendix A.3.

## 5 CONCLUSION

LucidFlux demonstrates that caption-free photo-realistic image restoration is best achieved by *when, where, and what* to condition a large diffusion transformer, rather than by adding parameters or prompts. A lightweight dual-branch conditioner—grounded in the degraded input and a lightly restored proxy—and a timestep- and layer-adaptive modulation schedule recover high-frequency detail while preserving global structure and suppressing artifacts, all with a frozen Flux.1 backbone. SigLIP-based semantics provide training–inference consistency without captions. To make post-training practical, we introduce, to our knowledge, the first publicly documented and extensively validated IR data-filtering pipeline. It is fully automatic once hyper-parameters are fixed and scales to 342K high-quality images and 1.36M paired samples, supplying structure-rich supervision at the capacity needed by large DiTs. Across real and synthetic benchmarks, LucidFlux delivers state-of-the-art perceptual quality and semantic fidelity with competitive runtime and minimal trainable overhead. We hope the pipeline, data recipe, and design insights provide a reliable foundation for restoration in the wild, and inspire future work on learned data selection, multi-frame/video extensions, and higher-resolution backbones—all while retaining caption-free inference.

**Acknowledgments.** This work is supported by the Guangdong Science and Technology Department (No. 2024ZDZX2004), the Nansha Key Area Science and Technology Project (No. 2024ZD006), and the National Natural Science Foundation of China (Project No.82572383).

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

## A  APPENDIX

### A.1  LIKELIHOOD OF DEGRADATION-RELATED TERMS IN CAPTIONS GENERATED BY DIFFERENT MULTIMODAL LARGE LANGUAGE MODELS

When using captions from VLM as semantic guidance for restoration tasks, a potential risk is that these models may unintentionally introduce degradation-related terms (e.g., blur, noise, or low resolution). Such bias can mislead the restoration model by attributing degradations to input images even when they are not visually apparent. To quantify this effect, we evaluate the occurrence of degradation-related descriptions in captions generated by a set of representative VLMs on RealLQ250, specifically LLaVA-v1.6-Vicuna-13B (Liu et al. (2024)) and Qwen2.5-VL-7B-Instruct (Team (2025)). Each caption is produced using the same prompt as DreamClear (Ai et al. (2024)), i.e., *"describe the key subjects and style"*, which is designed to neutrally guide the model toward content description without explicitly emphasizing or suppressing degradation cues. We then employ **Gemini-2.5-Flash-Image** as an external evaluator to analyze whether captions contain degradation-related mentions. Each caption is processed with a structured instruction to extract and categorize any degradation-related terms.

---

**Prompt A.1 (Identifying Quality Degradations in Image Captions)**

You are a professional image quality analysis expert. Carefully analyze the following image description text and identify any image quality issues that are either explicitly mentioned or implicitly implied.
Image description text: {caption_content}
Your task is to identify quality issues mentioned in the description. Focus on:
- Sharpness issues such as blur, unclear details, or defocus
- Noise, grain, or artifacts
- Low resolution, compression traces, or general quality problems
- Overexposure, underexposure, or color distortion
- Physical damage such as scratches, stains, or aging
Return results strictly in the following JSON format, without any additional explanation or text:

```
{
    "caption_content": "{caption_content}",
    "degradation_keywords": ["Extracted degradation-related terms"]
    ,
    "degradation_categories": {
        "Blur-related": ["blur", "unclear", "defocus"],
        "Noise-related": ["noise", "grain", "artifacts"],
        "Quality-related": ["resolution", "compression"],
        "Exposure-related": ["overexposure", "underexposure", "
    color issues"],
        "Damage-related": ["damage", "stains", "aging"]
    },
    "degradation_score": 0.0,
    "severity_level": "None/Minor/Moderate/Severe",
    "primary_issues": ["Main issue types"],
    "analysis_summary": "Brief analysis summary"
}
```

Scoring standard:
- `degradation_score`: 0 means no degradation, $< 0.3$ minor, 0.3–0.6 moderate, $> 0.6$ severe
- If no quality issues are mentioned in the text, set all arrays empty, score = 0, severity_level = "None"
**Important:** Only return pure JSON format results, without markdown code blocks or extra commentary.

---

If a caption contains terms that explicitly refer to image degradations, such as blur, noise, low resolution, or compression artifacts, we consider it a *degradation-related* caption. Table 5 presents the likelihood of captions containing such degradation-related terms. We observe that LLaVA-v1.6-vicuna-13b produces degradation-related captions in 17% of cases, whereas Qwen2.5-VL-7B-Instruct exhibits a higher occurrence of 24%. This indicates that different VLMs vary in their tendency to

Table 5: Occurrence rates (%) of degradation-related terms in captions generated by different VLMs on RealLQ250.

| Model | LLaVA-v1.6-Vicuna-13B | Qwen2.5-VL-7B-Instruct |
|---|---|---|
| With Degradation (%) | 17 | 24 |

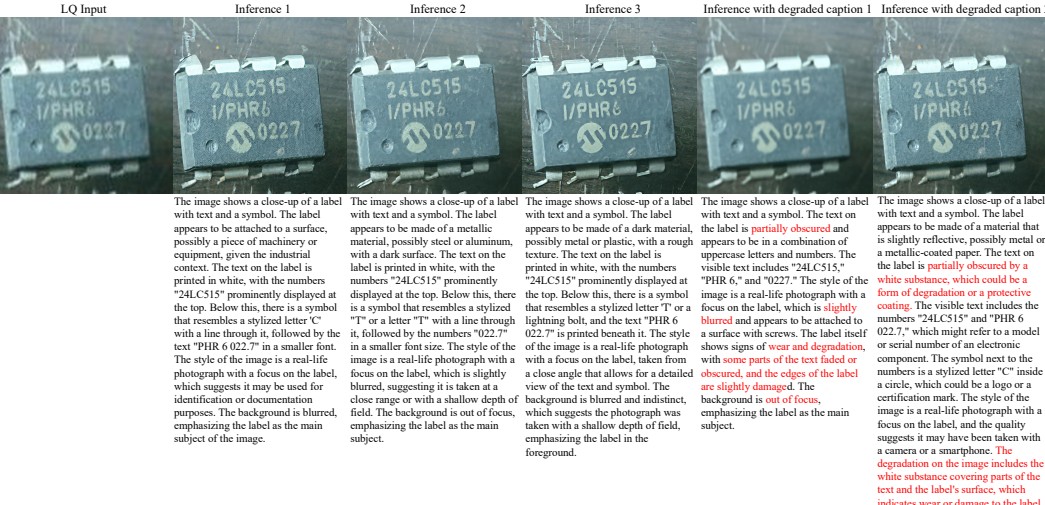

Figure 5: Impact of captions with and without degradation-related descriptions on restoration results. The second to fourth columns illustrate that inconsistent captions generated by the same VLM across different runs lead to variations in the restoration outcomes. The fifth and sixth columns show that captions containing explicit degradation descriptions misguide the restoration model and result in inferior quality compared with captions focusing purely on content and style.

introduce degradation cues into captions, which may potentially bias downstream restoration tasks if these captions are directly used as supervision.

## A.2 IMPACT OF DEGRADATION-RELATED CAPTIONS ON MODEL RESTORATION

To further investigate the influence of degradation-related descriptions in VLM-generated captions on restoration performance, we conducted experiments using two types of captions generated by LLaVA-v1.6-vicuna-13b. The first type uses the prompt *"Describe the key subjects and style"* to generate captions without emphasizing image degradations, while the second type uses the prompt *"Describe the key subjects and style, retain the descriptions of degradations on the image"* to produce captions that explicitly include degradation-related content.

As shown in Figure 5, two patterns emerge from the qualitative results. First, captions generated by the same VLM using the neutral prompt exhibit variability across multiple runs, resulting in differences in the restoration outputs for the same input image. Second, when captions explicitly include degradation-related descriptions, the model's restoration performance is adversely affected, producing outputs of lower perceptual quality compared with captions that focus solely on key subjects and style. These findings indicate that both the consistency and content of VLM-generated captions can significantly influence downstream restoration performance, underscoring the importance of controlling for degradation-related content when employing such captions as guidance.

These observations further highlight the practical limitations of relying on VLM-generated captions during inference. The variability in captions leads to inconsistent restoration results, the presence of degradation-related descriptions can mislead the model and reduce output quality, and generating captions introduces additional computational overhead. Together, these factors underscore the advantages of a caption-free approach, which avoids reliance on potentially inconsistent or misleading textual guidance while reducing inference cost and maintaining robust restoration performance.

Table 6: Ablation on caption usage under a fixed Flux.1 backbone on RealSR. GT captions are generated using high-quality reference images; VLM captions are generated from the low-quality inputs at inference time. Latency refers to additional inference overhead.

| Method | CLIP-IQA+ ↑ | Q-Align ↑ | MUSIQ ↑ | MANIQA ↑ | NIMA ↑ | CLIP-IQA ↑ | NIQE ↓ | Latency |
|---|---|---|---|---|---|---|---|---|
| GT caption (LLaVA on GT) | **0.7111** | **3.8713** | 70.1654 | **0.5686** | 5.0632 | **0.7032** | 4.6775 | +10.426s |
| VLM caption (LLaVA on LRP) | 0.7060 | 3.8168 | 69.4371 | 0.5533 | 5.0475 | 0.7030 | 4.6249 | +10.057s |
| Ours (caption-free) | 0.7074 | 3.7555 | **70.2005** | 0.5437 | **5.1072** | 0.6783 | **4.2893** | **+0.012** |

Table 7: Comparison of different backbone update strategies on RealLQ250.

| Training Strategy | CLIP-IQA ↑ | NIQE ↓ | MUSIQ ↑ | Memory |
|---|---|---|---|---|
| Attention-only Fine-Tuning | 0.654 | **3.52** | 66.27 | 60.30 GB |
| Full Fine-Tuning | 0.594 | 3.796 | 63.30 | 76.16 GB |
| Ours | **0.7122** | 3.6742 | **73.01** | 76.53 GB |

## A.3 ABLATION ON CAPTION USAGE FOR A CLEAN ATTRIBUTION OF THE CAPTION-FREE DESIGN

To isolate the effect of our caption-free formulation, we evaluate three variants that share the same Flux.1 backbone and differ only in their use of captions: (i) captions generated by LLaVA from the ground-truth reference image, (ii) captions inferred from the lightly restored proxy as used at test time, and (iii) our caption-free variant. As shown in Table 6, both caption-based settings introduce $\sim 10$ seconds per image due to VLM processing. Moreover, even the idealized GT captions do not translate into superior image quality—the caption-free model achieves the best MUSIQ (70.2005), NIMA (5.1072), and NIQE (4.2893), while VLM captions further degrade performance across all metrics. These results confirm that performance improvements do not stem from captions but from a more robust caption-free design, which also avoids substantial inference latency and external dependencies.

## A.4 COMPARISON OF BACKBONE UPDATE STRATEGIES

We compare different strategies for updating the DiT backbone, including attention-only updates, full fine-tuning of all parameters, and our frozen-backbone adaptation. As shown in Table 7, updating only the attention layers results in stable training but reaches a performance ceiling (CLIP-IQA 0.654 / MUSIQ 66.27), suggesting that adapting cross-attention alone is insufficient for handling real-world degradations. Full fine-tuning increases the model's adaptability but can cause catastrophic forgetting, leading to reduced perceptual quality (CLIP-IQA 0.594 / MUSIQ 63.30). Our approach obtains the best perceptual performance (CLIP-IQA 0.7122 / MUSIQ 73.01) while maintaining stable optimization. Although its memory footprint is similar to full fine-tuning, it avoids destabilizing the generative prior and therefore offers a more favorable balance between performance and training reliability.

## A.5 ADDITIONAL RELATED WORKS

**Large-Scale Image Restoration Datasets.** The availability of large, high-quality datasets is critical for training generative restoration models Chen et al. (2024; 2025); Lin et al. (2025b). Existing datasets exhibit notable limitations: LSDIR (Li et al. (2023)) provides 85K images but depends on manual filtering, SUPIR (Yu et al. (2024)) collects 20M images without disclosing quality control procedures, and DreamClear (Ai et al. (2024)) generates 1M images via SDXL fine-tuning at a cost of 1280 V100 GPU days. To overcome these constraints, LucidFlux employs a fully automated three-stage filtering pipeline integrating blur detection, flat-region detection, and perceptual quality assessment. This approach produces diverse, structurally rich datasets that are reproducible, scalable, and suitable for training billion-parameter diffusion backbones efficiently.

**Transformer-based T2I models (DiTs).** Recent text-to-image systems increasingly adopt Transformer backbones—either diffusion transformers (DiTs) or rectified-flow transformers (RFTs)—which scale well and capture long-range dependencies in latent space (Peebles & Xie

(2022); Ye et al. (2025); Chen et al. (2026); Lai et al. (2026)). Stable Diffusion 3 (SD3) introduces a *Multimodal Diffusion Transformer (MMDiT)* with *separate* weights for image and text tokens and bidirectional information flow; it is trained with *rectified flow* and improved noise sampling biased toward perceptually relevant scales, yielding stronger text comprehension and typography (Esser et al. (2024)). PixArt-$\alpha$ proposes an efficient DiT recipe—three-stage training (pixel dependency, text–image alignment, aesthetics), injecting cross-attention into DiT, and dense pseudo-captioning—achieving 1024px photorealistic quality at a fraction of typical compute (Chen et al. (2023a)). FLUX (Labs (2024)) scales a *rectified-flow Transformer* (rather than a diffusion transformer), with open-weight variants (e.g., `dev/schnell`) built around cross-attention over text embeddings. Building on this line, **LucidFlux** leverages a large MM-DiT backbone (Flux.1) and specializes conditioning for caption-free restoration, improving detail fidelity while preserving semantics.

## A.6 EXTENDED DATASET ANALYSIS

To further examine semantic diversity, we visualize the CLIP image–text embeddings using t-SNE. We randomly sample 10K images from our filtered data, while using all available images from Flickr2K and DIV2K. As shown in Figure 6, our dataset spans a substantially broader semantic range, reflecting richer and more diverse image–text concepts. This confirms the advantage of our dataset in supporting models that rely on wide semantic generalization.

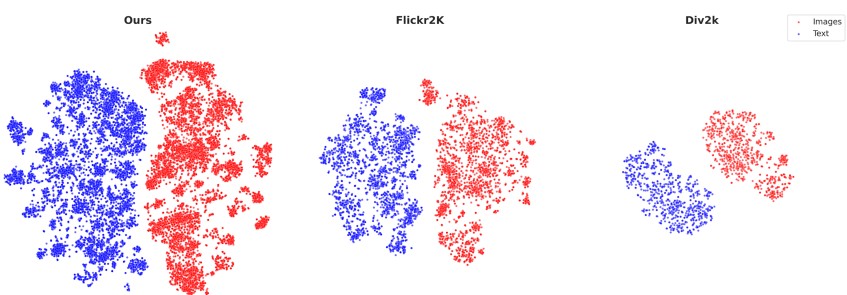

Figure 6: t-SNE visualization of CLIP image–text embeddings. Our dataset covers a broader semantic range than Flickr2K and DIV2K, indicating richer image–text diversity.

### A.6.1 ANALYSIS OF FILTERING SETTINGS AND SEMANTIC COVERAGE

In this section, we report the sample sizes resulting from different filtering configurations and analyze their corresponding semantic coverage. The filtering pipeline is applied sequentially in the following order: blur detection, flat-region detection, and IQA filtering.

For the baseline setting, we first apply blur detection by retaining images with Laplacian variance in the range $150 \leq S_{\text{blur}}(I) \leq 8000$. Next, flat-region detection excludes images containing more than 50% textureless patches, where a patch is considered textureless if its gradient variance satisfies $S_{\text{flat}} = \text{Var}\left(\sqrt{(\partial_x I)^2 + (\partial_y I)^2}\right) < 800$. Finally, we retain the top 20% of images according to their CLIP-IQA scores. After applying these three steps, 234,287 samples remain in the baseline configuration.

Table 8 summarizes the number of retained samples for each threshold configuration. It can be observed that stricter thresholds (e.g., Blur Variant (Set 1), Flat Variant (Set 3), and CLIPIQA Variant (Set 5)) reduce the sample size, whereas more relaxed thresholds (e.g., Blur Variant (Set 2), Flat Variant (Set 4), and CLIPIQA Variant (Set 6)) increase it.

**Semantic Coverage Analysis.** To evaluate semantic diversity and coverage under different conditions, we randomly sample 1k, 5k, and 10k images from each configuration and visualize their embeddings using t-SNE, as shown in Fig. 7. The results demonstrate that under the same 10k sample size, different filtering settings yield largely similar semantic coverage. As shown in the t-SNE visualization, the distributions of baseline and variant settings (Blur Variant, Flat Variant, and CLIPIQA Variant) exhibit only minor differences in semantic diversity, indicating that threshold

Table 8: Sample Size After Filtering for Different Settings

| Setting | Threshold | Sample Number |
|---|---|---|
| Pool | None | 2,900,747 |
| Our Baseline | Blur 150-8000, Flat 800, 50%, CLIPIQA 20% | 234,287 |
| Blur Variant (Set 1) | Blur 200-6000, Flat 800, 50%, CLIPIQA 20% | 219,966 |
| Blur Variant (Set 2) | Blur 100-10000, Flat 800, 50%, CLIPIQA 20% | 245,090 |
| Flat Variant (Set 3) | Blur 150-8000, Flat 1000, 40%, CLIPIQA 20% | 163,482 |
| Flat Variant (Set 4) | Blur 150-8000, Flat 600, 60%, CLIPIQA 20% | 301,637 |
| CLIPIQA Variant (Set 5) | Blur 150-8000, Flat 800, 50%, CLIPIQA 10% | 123,007 |
| CLIPIQA Variant (Set 6) | Blur 150-8000, Flat 800, 50%, CLIPIQA 30% | 330,618 |

Table 9: Image quality metrics under different filtering settings. Higher is better for all metrics except NIQE, where lower is better.

| Setting | CLIP-IQA+ ↑ | Q-Align ↑ | MUSIQ ↑ | MANIQA ↑ | NIMA ↑ | CLIP-IQA ↑ | NIQE ↓ |
|---|---|---|---|---|---|---|---|
| Our Baseline | **0.7406** | **4.3935** | **73.0088** | **0.5589** | 5.4836 | **0.7122** | 3.6742 |
| Blur Var. (Set 1) | 0.7023 | 4.3041 | 71.3748 | 0.5057 | 5.4614 | 0.6622 | 3.4348 |
| Blur Var. (Set 2) | 0.7072 | 4.2464 | 71.8887 | 0.5140 | **5.5129** | 0.6590 | 3.8099 |
| Flat Var. (Set 3) | 0.7011 | 4.2164 | 71.3910 | 0.5161 | 5.4096 | 0.6573 | 3.6507 |
| Flat Var. (Set 4) | 0.6913 | 4.2795 | 70.2861 | 0.4973 | 5.4570 | 0.6584 | **3.3471** |
| CLIPIQA Var. (Set 5) | 0.6749 | 3.9777 | 69.8807 | 0.4827 | 5.2303 | 0.6226 | 3.5555 |
| CLIPIQA Var. (Set 6) | 0.6897 | 4.2487 | 70.3159 | 0.4891 | 5.3737 | 0.6418 | 3.4848 |

variations have limited impact on the overall semantic coverage when the sample size is fixed. On the other hand, under the same baseline setting, increasing the sample size from 1k to 5k to 10k leads to a clear expansion of semantic coverage. The t-SNE plots show that as the sample size grows, the clusters broaden substantially, covering a wider range of the semantic space. This demonstrates that sample size plays a more significant role than specific threshold settings in enhancing the diversity and robustness of the dataset.

These findings confirm the stability and effectiveness of our data curation strategy, showing that semantic coverage remains consistent across different filtering configurations when sample size is fixed, while being substantially improved by increasing the number of samples.

### A.6.2 IMPACT OF FILTERING SETTINGS ON IMAGE QUALITY METRICS

We evaluate the resulting datasets under each filtering configuration using seven no-reference image quality assessment metrics: CLIP-IQA+, QAlign, MUSIQ, MANIQA, NIMA, CLIP-IQA, and NIQE. As summarized in Table 9, our baseline configuration yields the highest scores across nearly all metrics, confirming its effectiveness in selecting high-quality, visually coherent images.

Stricter thresholds—such as Blur Variant (Set 1), Flat Variant (Set 3), or CLIPIQA Variant (Set 5)—generally reduce quality scores, particularly in QAlign and MUSIQ, which are sensitive to semantic clarity and structural detail. For example, selecting only the top 10% by CLIP-IQA (Set 5) lowers QAlign from 4.39 to 3.98 and MUSIQ from 73.01 to 69.88, indicating a loss of diverse but still high-quality content. Conversely, relaxed thresholds—like Flat Variant (Set 4)—admit more texture-poor or blurry images, resulting in decreased MANIQA (0.4973) and higher NIQE (3.35), reflecting reduced perceptual fidelity.

Notably, even when CLIP-IQA is not the evaluation metric, the baseline maintains strong performance across all indicators, suggesting that the multi-stage filtering strategy enhances overall image quality beyond the selection criterion alone. These findings, together with the semantic coverage analysis, demonstrate that the baseline offers a well-calibrated trade-off between data quality, diversity, and scale.

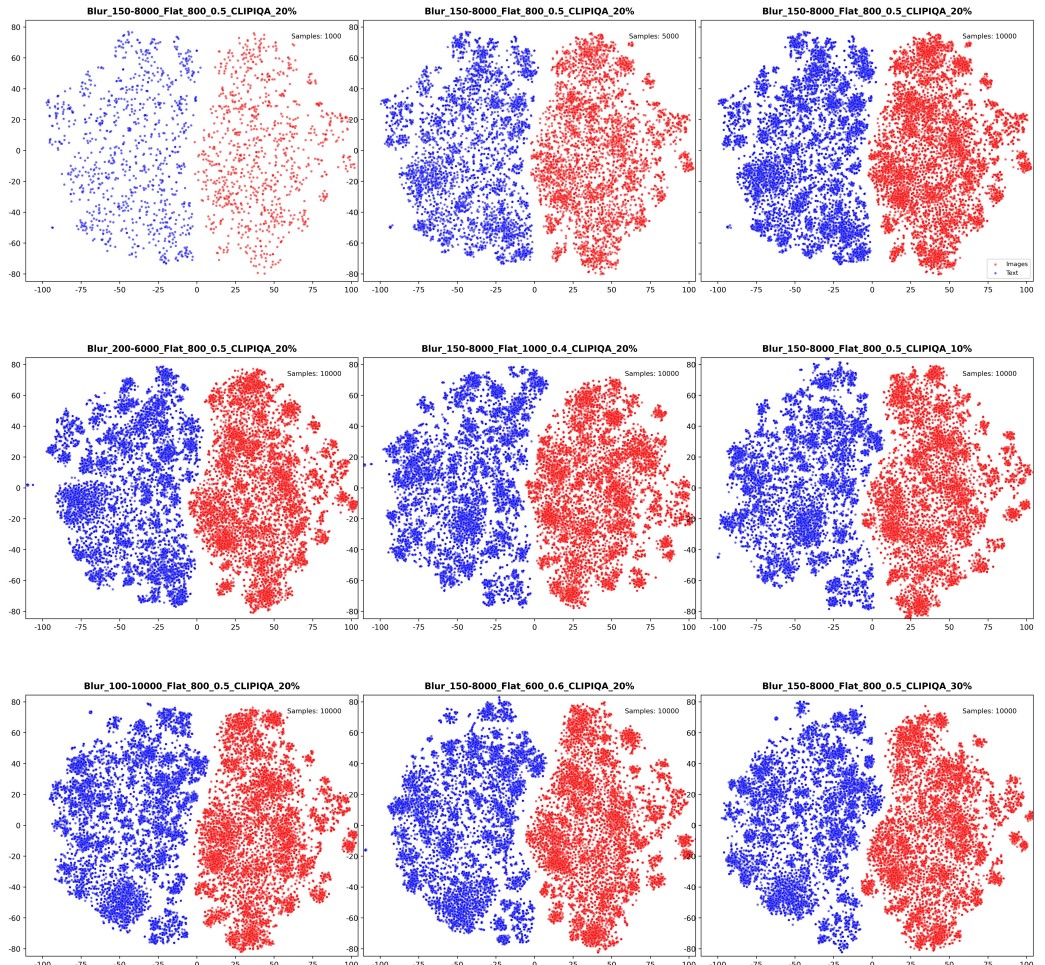

Figure 7: t-SNE visualization of semantic coverage under different configurations. The first row displays the baseline setting with sample sizes of 1k, 5k, and 10k (left to right). The second and third rows show variant settings at 10k samples: Blur Variant Set 1 (2nd row, 1st col), Set 2 (3rd row, 1st col); Flat Variant Set 3 (2nd row, 2nd col), Set 4 (3rd row, 2nd col); CLIPIQA Variant Set 5 (2nd row, 3rd col), Set 6 (3rd row, 3rd col).

## A.7  EXTENDED VISUAL COMPARISONS

To provide a more comprehensive evaluation, we present extended qualitative results across all benchmark datasets. Figures 9–14 include representative examples from RealLQ250, DRealSR, RealSR, DIV2K-Val, and LSDIR-Val. These comparisons consistently demonstrate that our method produces sharper edges, more faithful textures, and better preservation of semantic structures compared with existing open-source state-of-the-art approaches. The additional results further corroborate the advantages of our approach observed in the main paper.

## A.8  VISUAL COMPARISON WITH CLOSE-SOURCE COMMERCIAL METHODS

Figure 8 illustrates representative visual results on RealLQ250. HYPIR-FLUX and Seedream 4.0 fail to fully remove degradations, leaving noticeable residual artifacts. Topaz suppresses degradations more effectively but generates flat and over-smoothed textures. Gemini-NanoBanana provides visually plausible outputs but often struggles to recover high-frequency details. MeiTu SR shows relatively strong restoration ability, producing sharper and more natural results compared with most commercial counterparts. Among the evaluated models, LucidFlux consistently delivers

the sharpest structures and most faithful details, particularly in fine-grained regions, while maintaining high structural fidelity and reliability.

## A.9 INFERENCE DETAILS

For all experiments, we use the FlowMatch Euler sampling introduced for SD3's rectified-flow formulation and implemented in Diffusers' FlowMatchEulerDiscreteScheduler (Esser et al. (2024)) to sample FLUX (Labs (2024)) and inherit the adaptive shift adjustments from the official Flux implementation, ensuring stable sampling dynamics and consistent step-wise updates. All inference is performed with 28 sampling steps in FP16 precision and utilizes the wavelet color alignment method fro (Ai et al. (2024)) and the full default instruction is restore this image into high-quality, clean, high-resolution result.

## A.10 LIMITATIONS

While LucidFlux attains strong perceptual quality and semantic fidelity, several practical limitations remain:

**Large model scale.** LucidFlux is built on a high-capacity DiT backbone (Flux.1). This provides rich generative priors but entails substantial parameter count and compute cost. Training generally requires multi-GPU setups; for inference, high-end GPUs are preferable to maintain reasonable throughput.

**Inference GPU memory.** VRAM usage during inference is sizable and grows with input resolution and batch size. Transformer-based diffusion exhibits quadratic attention complexity with respect to token count, so higher resolutions can quickly amplify memory pressure. This constrains deployment on memory-limited devices unless tiling or resolution reductions are used.

**Sampling steps vs. quality.** High-quality outputs typically require more than ~15 denoising steps. Fewer steps may lead to over-smoothing and loss of fine textures. This introduces a latency–quality trade-off that can be restrictive for real-time or interactive applications.

**Mitigations.** Promising directions include model compression (distillation/pruning), low-precision inference, memory-efficient attention, and step reduction via progressive distillation. These optimizations are orthogonal to our method and could reduce compute and memory while preserving quality.

## A.11 ACKNOWLEDGMENTS

We thank the maintainers and communities behind Flux.1 (Labs (2024)), PixArt-$\alpha$ (Chen et al. (2023a)), SD/SDXL (Rombach et al. (2022); Podell et al. (2023)), SigLIP, and the dataset providers for LSDIR, DIV2K, Flickr2K, RealSR, and DRealSR for making their resources available to the research community.

**Intended application scope.** LucidFlux is designed for benign restoration of non-sensitive, natural photographic images with mixed, unknown degradations (e.g., sensor noise, motion blur, compression). Appropriate uses include consumer photo enhancement, archival preservation, academic research, and benchmarking under unknown degradations. The model aims to improve perceptual quality while preserving semantics, but as a generative diffusion system it may synthesize plausible details not present in the input.

**Out-of-scope scenarios.** The method is *not* intended for domains that require pixel-accurate fidelity or expert supervision (e.g., medical imaging, scientific microscopy, satellite/remote sensing, or legally binding forensic evidence). It is also not tailored for document restoration or OCR-critical text recovery.

**Prohibited or discouraged uses.** LucidFlux should not be used to circumvent privacy, safety, or consent—such as deblurring or enhancing faces, license plates, or personally identifiable content for surveillance or re-identification; removing watermarks or intentional obfuscation; fabricating or altering imagery for deception; or generating identity-sensitive content (e.g., deepfakes). When restoration might affect downstream decisions about people, human oversight is required.

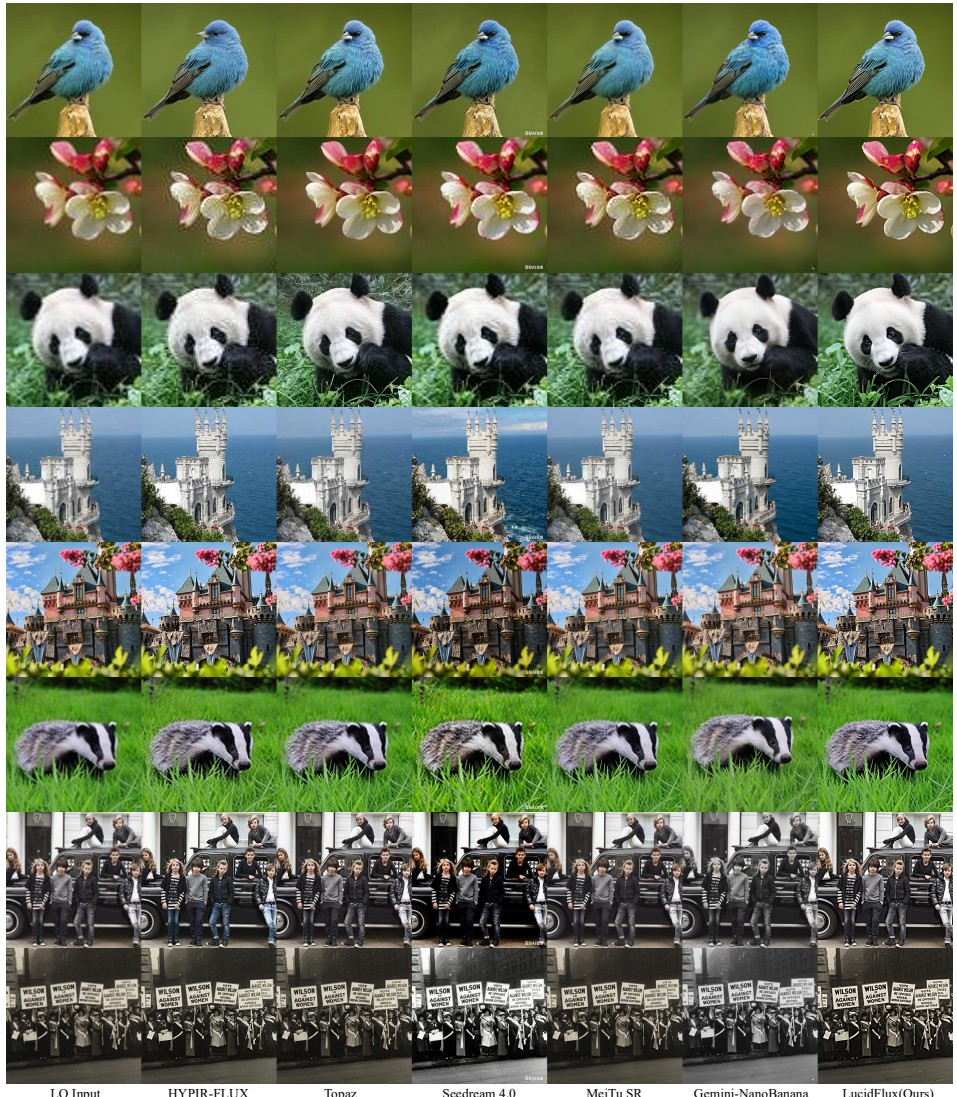

LQ Input    HYPIR-FLUX    Topaz    Seedream 4.0    MeiTu SR    Gemini-NanoBanana    LucidFlux(Ours)

Figure 8: Qualitative comparison with commercial models on RealLQ250.

**Operational caveats.** Because performance depends on degradation type and sampling steps, outputs should be reviewed before downstream use, especially in safety-critical or regulatory contexts. If exact visual truth is required, classical reconstruction baselines or domain-specific methods with uncertainty quantification are preferable.

## A.12 LLM USAGE

We used large language models solely for editorial assistance—to refine grammar and phrasing, improve clarity and flow, and condense overly verbose passages. No ideas, methods, code, figures, citations, or results were generated by an LLM, and no unverifiable content was introduced. All technical content, study design, experiments, analyses, and conclusions were conceived, executed, and validated by the authors, who take full responsibility for the manuscript.

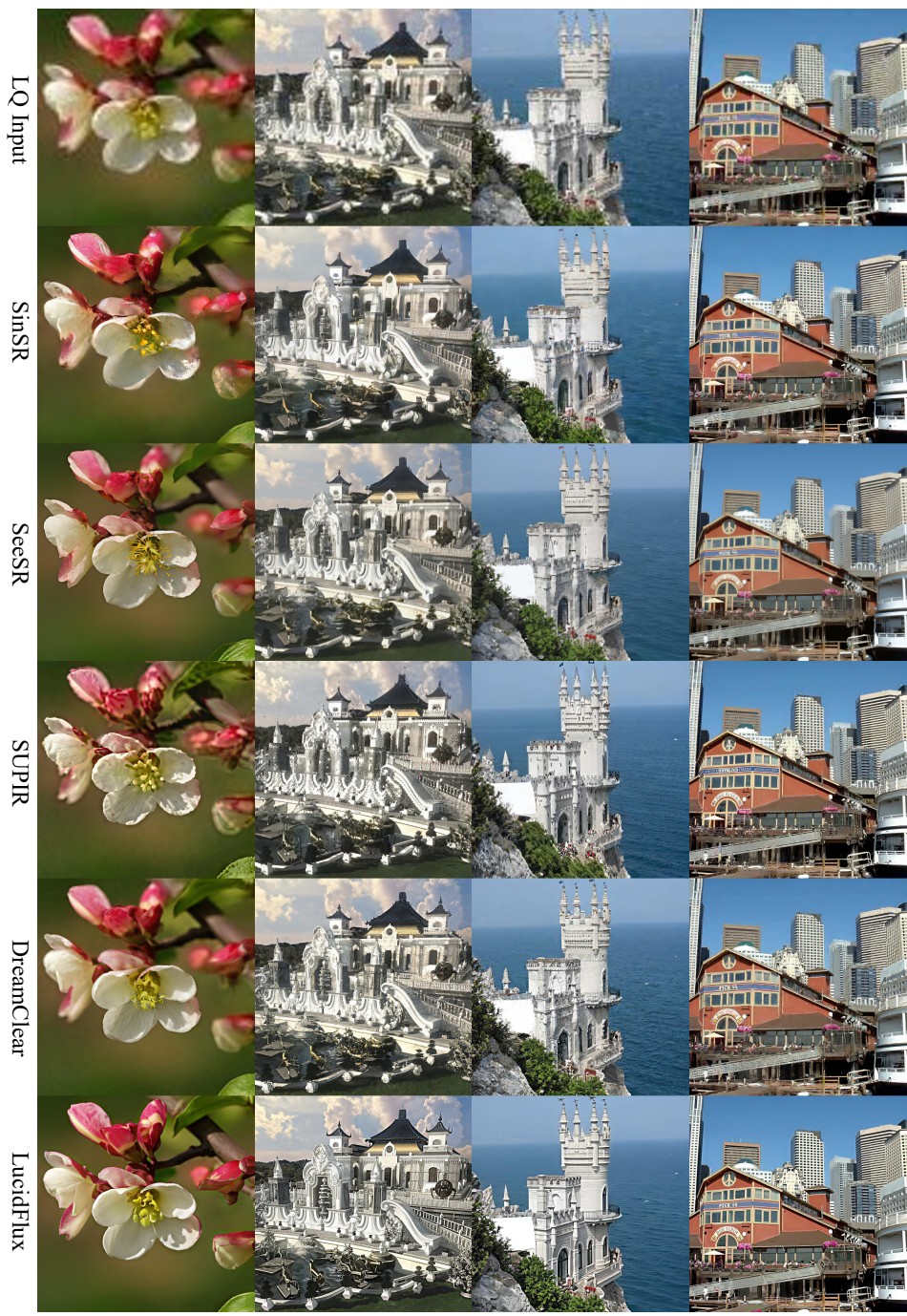

Figure 9: More examples of visual comparison with open-source state-of-the-art methods on Re-alLQ250.

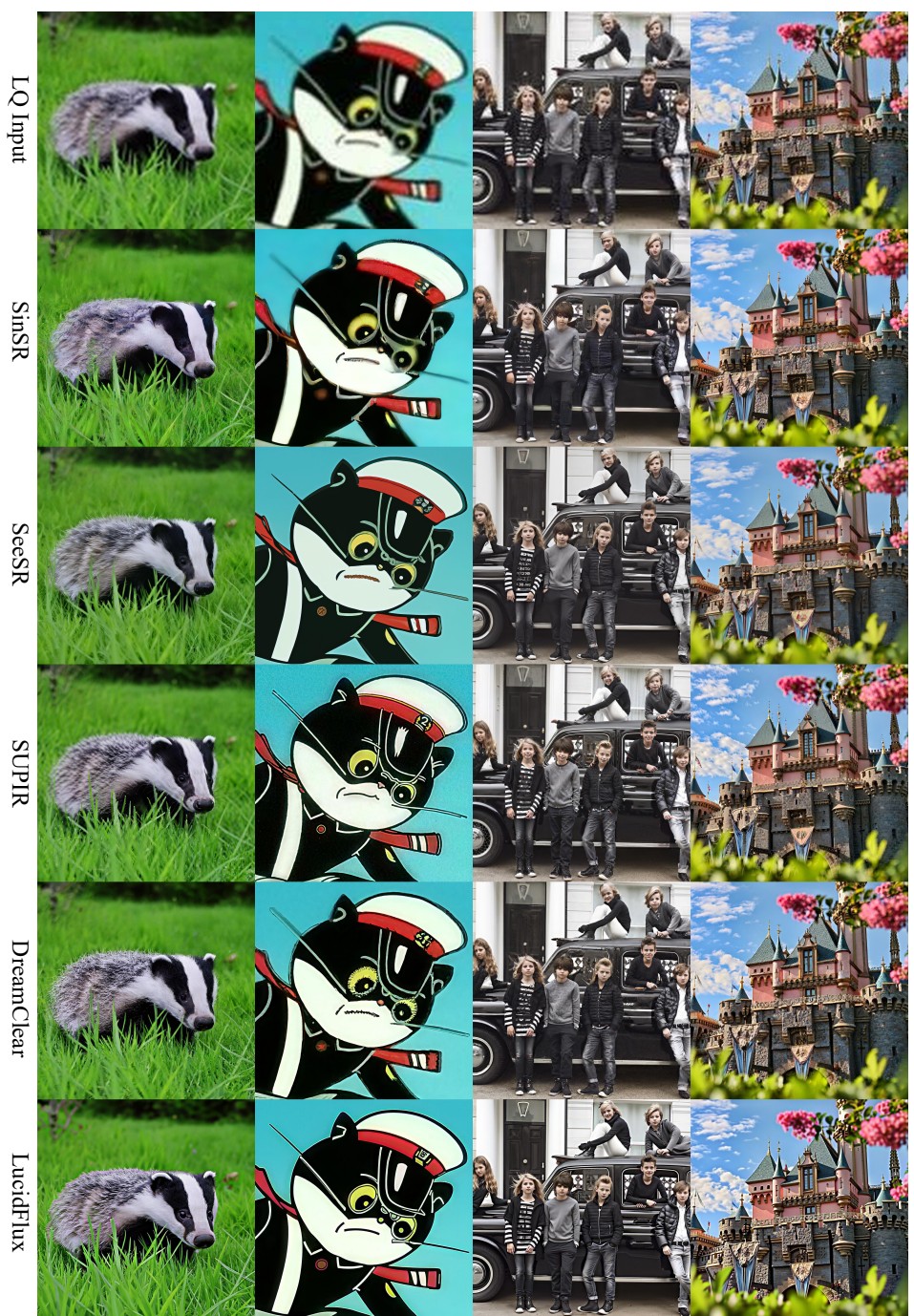

Figure 10: More examples of visual comparison with open-source state-of-the-art methods on Re-alLQ250.

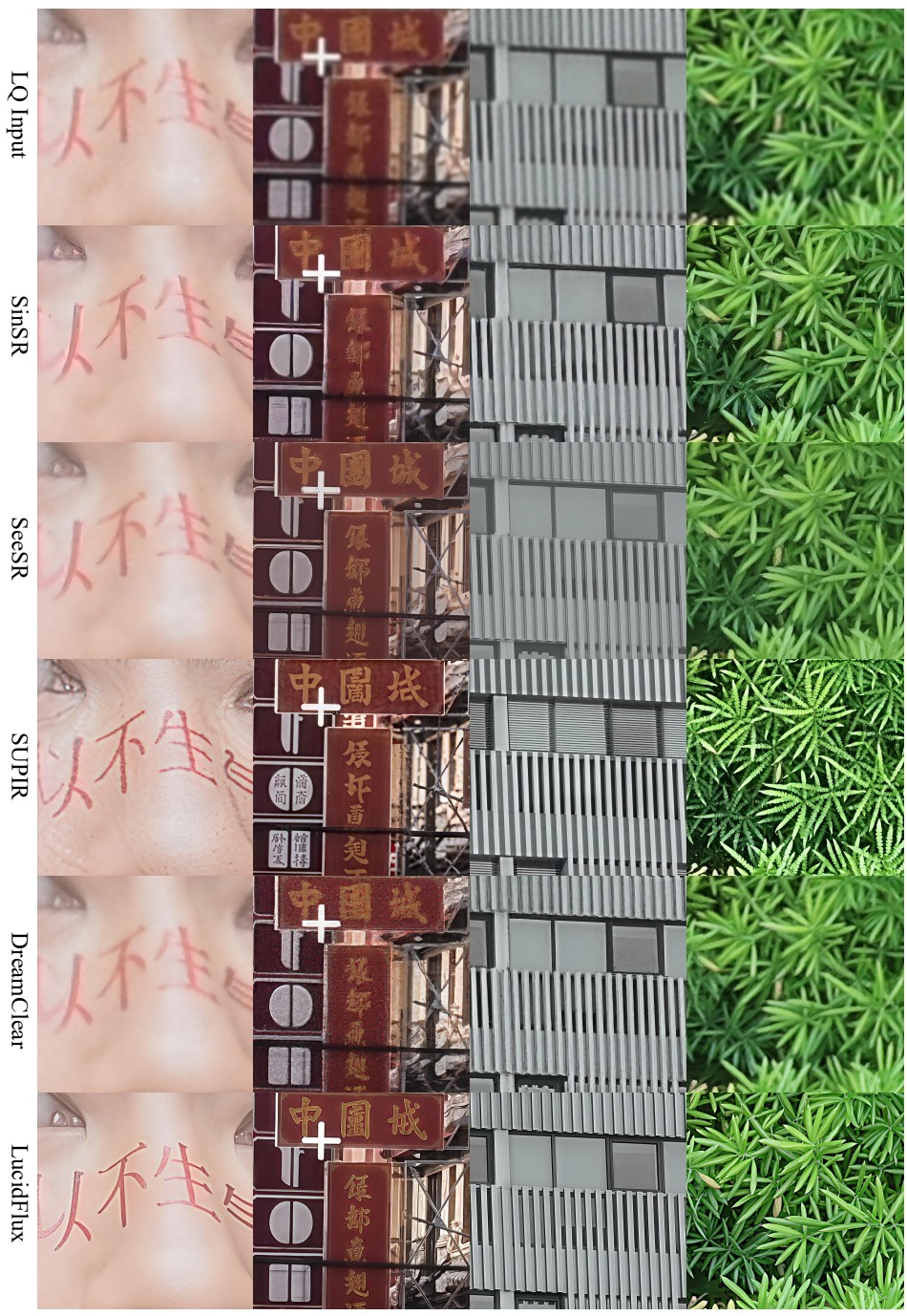

Figure 11: More examples of visual comparison with open-source state-of-the-art methods on DRealSR.

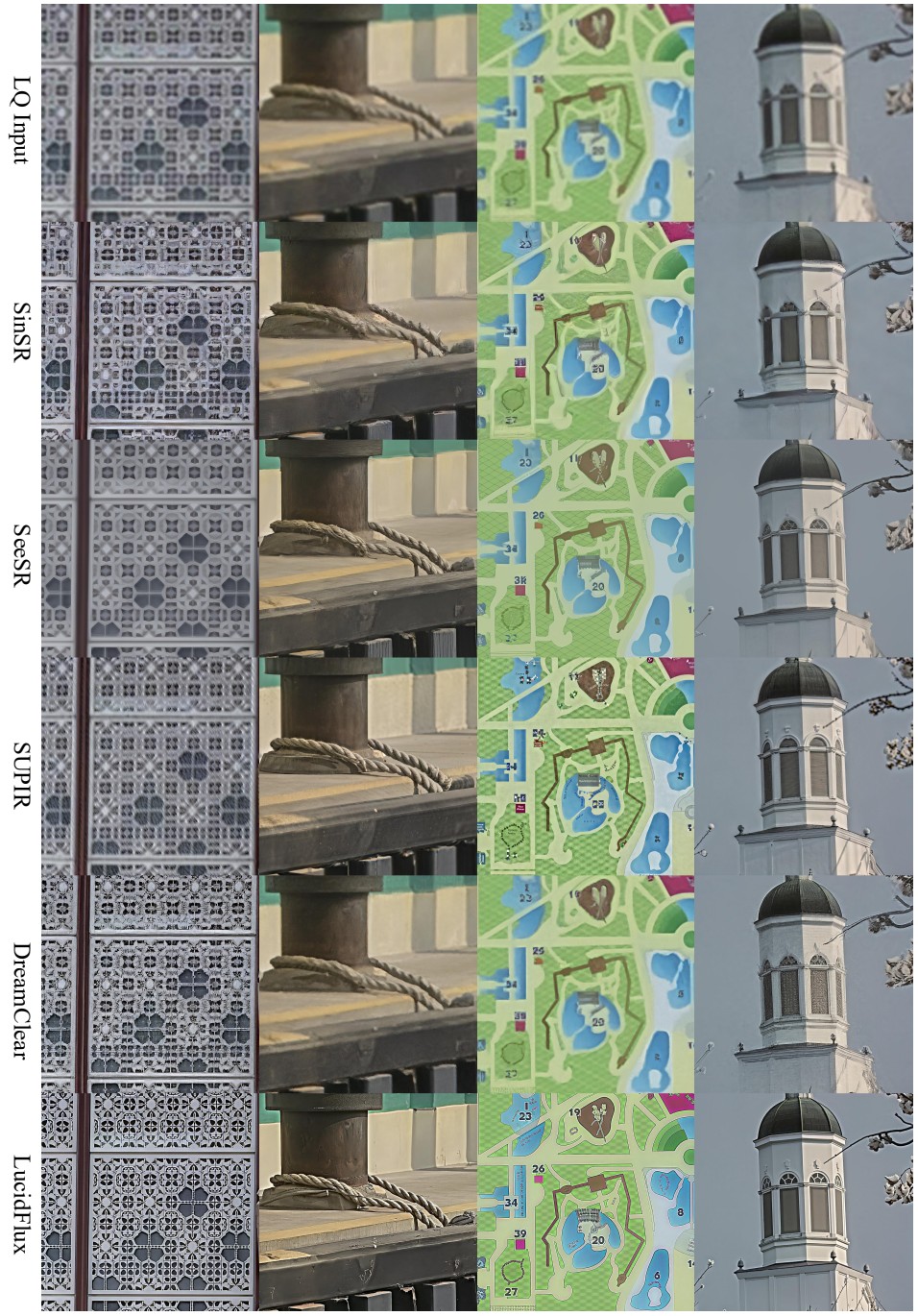

Figure 12: More examples of visual comparison with open-source state-of-the-art methods on RealSR.

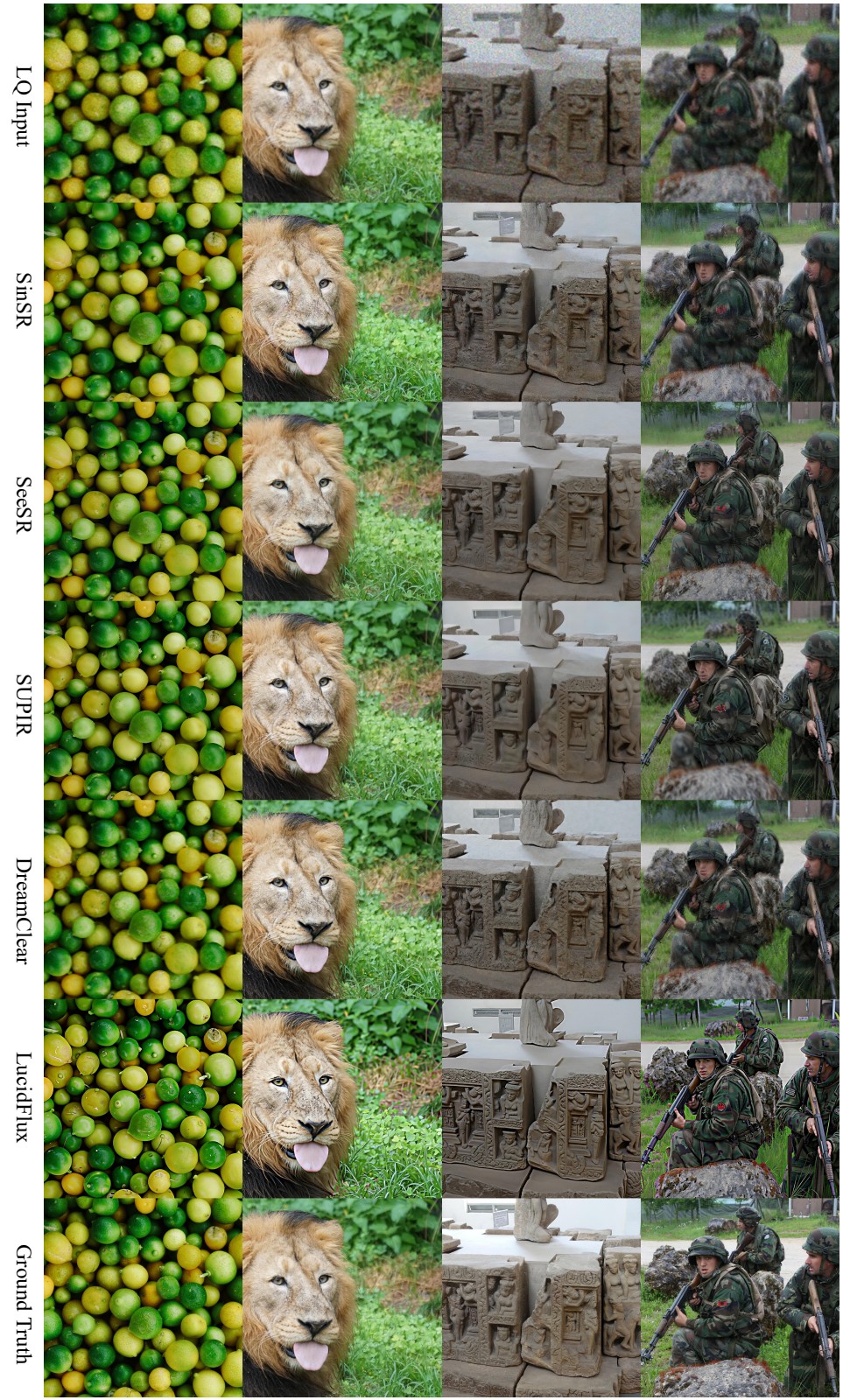

Figure 13: More examples of visual comparison with open-source state-of-the-art methods on Div2k-Val.

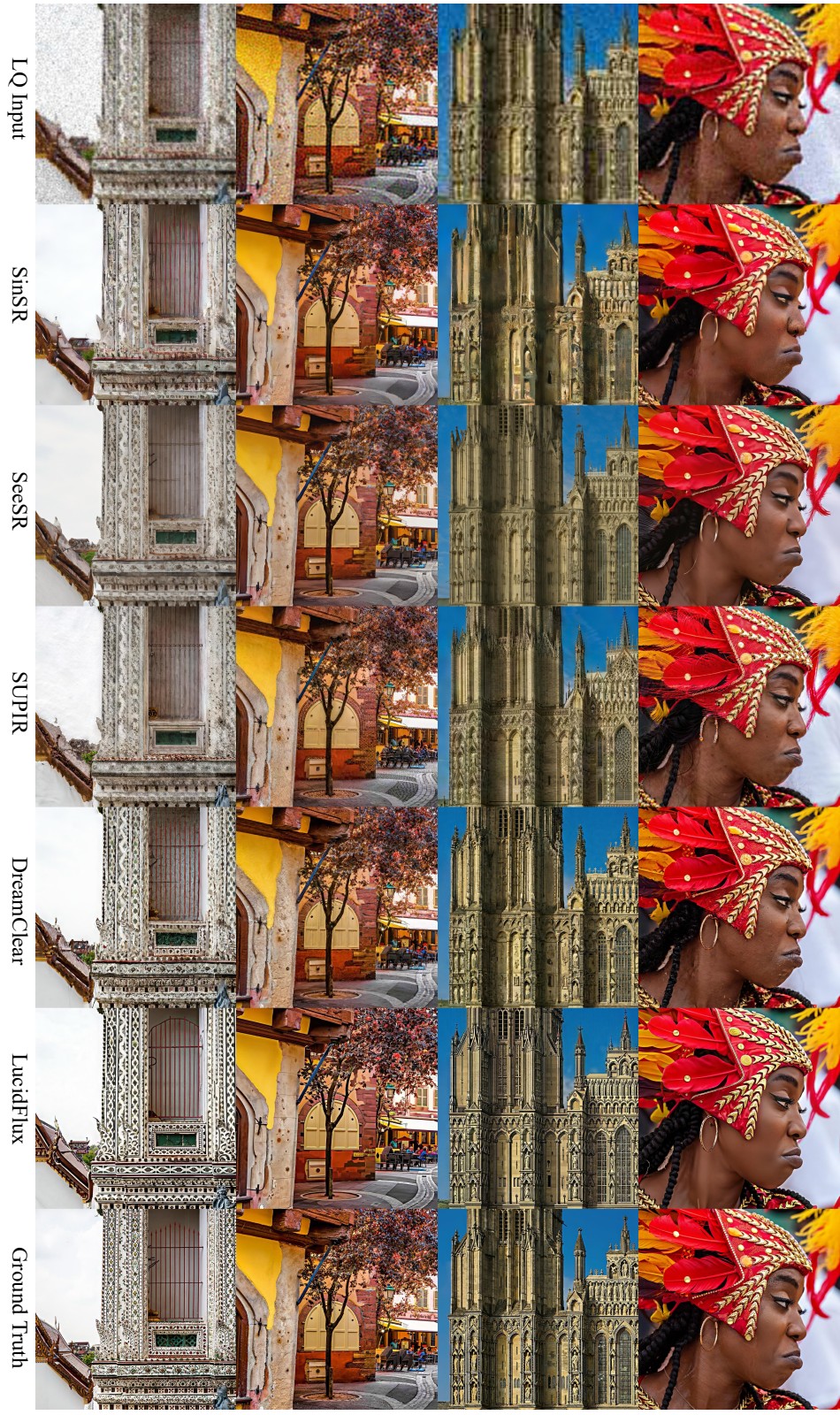

Figure 14: More examples of visual comparison with open-source state-of-the-art methods on LSDIR-Val.

