# OpenReview forum: "LucidFlux: Caption-Free Photo-Realistic Image Restoration via a Large-Scale Diffusion Transformer"
_ICLR.cc/2026/Conference — ICLR 2026 Poster_

### Official Review · Reviewer_8VrS · 2025-10-27

**Soundness:** 3
**Presentation:** 2
**Contribution:** 2
**Rating:** 4
**Confidence:** 5

**Summary:**

This paper proposes LucidFlux, an image restoration framework that adapts a large diffusion transformer (Flux.1) for restoring images degraded by unknown mixtures, without relying on captions. The key contributions include a lightweight dual-branch conditioner that anchors geometry and suppresses artifacts by injecting signals from both the degraded input and a lightly restored proxy. Additionally, a timestep- and layer-adaptive modulation schedule enables context-aware, coarse-to-fine restoration. To ensure semantic alignment without text prompts, LucidFlux leverages SigLIP features from the proxy. The framework also introduces a scalable data curation pipeline for structure-rich supervision. Together, these innovations enable robust and efficient restoration with minimal parameter overhead.

**Strengths:**

1. The proposed method is simple and easy to follow.
2. The proposed method achieves state-of-the-art performance on real-world super-resolution datasets.

**Weaknesses:**

1. The usage of the term "Universal image restoration" is not accurate. In the low-level vision field, "Universal image restoration" usually refers to all-in-one image restoration tasks [1,2,3] in most papers. However, in my opinion, this paper mainly addresses real-world image restoration tasks.

2. In terms of model architecture, the proposed method adopts an identical dual-branch design as DreamClear, showing limited structural innovation.

3. I believe the proposed dataset is an important contribution of this paper. However, it is unclear whether it will be open-sourced.

4. The experiments comparing the proposed method with commercial models are not meaningful, as many of the models used for comparison are not super-resolution models.

5. This paper uses FLUX as the base model, which has significantly more parameters and computational cost compared to previous works. The authors need to provide inference efficiency comparisons with prior methods.


 References

[1] Controlling Vision-Language Models for Universal Image Restoration. ICLR, 2024.

[2] Universal Image Restoration Pre-training via Degradation Classification. ICLR, 2025.

[3] Selective Hourglass Mapping for Universal Image Restoration Based on Diffusion Model. CVPR, 2024.

**Questions:**

Refer to Weakness

---

> ### Author Response · Authors · 2025-11-26
> **Reply to Reviewer  8VrS Weakness 1-3**
>
> **We thank Reviewer 8VrS for the careful and constructive review. We appreciate the positive comments that the method is simple and easy to follow, and that LucidFlux achieves state-of-the-art performance on real-world super-resolution benchmarks with very strong visual quality. Below we address the reviewer’s concerns on terminology, architectural novelty, dataset contribution, commercial baselines, and efficiency.**
>
> ---
>
> ### Weakness 1 – Use of the term “Universal Image Restoration”
>
> We thank the reviewer for pointing this out. We agree that in the low-level vision literature **“Universal Image Restoration (UIR)” is commonly used to denote all-in-one restoration**, where a single model jointly handles denoising, deblurring, dehazing, SR, etc. [1,2,3]. In contrast, our work focuses on **photo-realistic restoration of real-world images under unknown mixed degradations** (e.g., noise, blur, compression, low resolution) without explicitly modeling each degradation type as a separate task.
>
> To avoid overstating the scope and to align with SUPIR [4], we now refer to our title as **“Caption-Free Photo-Realistic Image Restoration via a Large-Scale Diffusion Transformer”** rather than “Caption-Free Universal Image Restoration via a Large-Scale Diffusion Transformer”. Concretely, our goal is to recover **visually faithful, photo-realistic images from real-world degraded images**, rather than to unify all restoration tasks in one model.
>
> ---
>
> ### Weakness 2 – Dual-branch architecture vs. DreamClear
>
> We appreciate the reviewer’s concern about architectural novelty and fully acknowledge that we **adopt the dual-branch *paradigm*** introduced by DreamClear (structure anchoring vs. artifact suppression). However, LucidFlux is not an “identical” design; the key differences arise from **how the two branches are instantiated, injected, and scheduled in a large Flux.1 DiT**, and how they are integrated into a caption-free framework:
>
> 1. **Conditioner tailored to a 12B Flux.1 DiT.**
>
>     DreamClear attaches a heavy ControlNet-style dual branch on a relatively small backbone. Directly mirroring this design on a **12B Flux.1** model would be computationally prohibitive. LucidFlux instead uses a **latent-space-only, shallow, non-replicated conditioner** (a few MMDiT blocks per branch) that does *not* duplicate the main DiT. This keeps the overhead roughly constant with depth and makes dual-branch conditioning **practically scalable at 12B scale** while preserving the desired “geometry vs. artifact” decomposition.
>
> 2. **Timestep- and layer-adaptive modulation (TLCM).**
>
>     Rather than fusing branches in a static way, we introduce a **(timestep, layer)-aware gating mechanism**. A small modulation head takes normalized timestep and layer index and predicts per-channel scales and gates, so the relative influence of the LQ and LRP streams differs at early vs. late steps and shallow vs. deep layers. This explicitly exploits the temporal–hierarchical roles of DiTs and yields significant gains in our ablations (e.g., large improvements in MUSIQ and CLIP-IQA+ on RealLQ250).
>
> 3. **Caption-free SigLIP semantic pathway and curation pipeline.**
>
>     DreamClear relies on inference-time MLLM captions, whereas LucidFlux uses a **caption-free SigLIP-based semantic pathway** from a lightly restored proxy: one branch anchors structure and SigLIP-derived semantics, the other focuses on artifact removal on the LQ input. This is tightly coupled with our **IQA/structure-aware data curation pipeline** and removes MLLM latency and instability. This combination of caption-free semantics + lightweight dual-branch conditioner + TLCM is not present in DreamClear.
>
>
> Empirically, these design choices translate into **consistent improvements over DreamClear** across multiple IQA metrics on both synthetic and real-world benchmarks, while providing competitive or better end-to-end runtime (see also the efficiency discussion below).
>
> ---
>
> ### Weakness 3 – Dataset as a contribution and release plan
>
> We thank the reviewer for recognizing the importance of the dataset. Our curated dataset is constructed by filtering a **2.9M candidate pool** (Pexels/Unsplash + Photo-Concept-Bucket) with blur/flatness/IQA constraints to produce **structure-rich, high-quality HQ images**, and then synthesizing LQ counterparts via the Real-ESRGAN degradation pipeline. To the best of our knowledge, this is the **first large-scale, publicly reproducible photo-realistic restoration dataset** built from these sources with explicit structure/IQA filtering.
>
> We **will release** the curated HQ pool, the Real-ESRGAN degradation scripts, and the exact train/validation splits (subject to the original Pexels/Unsplash licenses) upon acceptance. This will allow other researchers to **reproduce our LQ–HQ pairs, benchmark new methods, and explore related restoration/SR tasks**, turning the dataset into a shared resource for the community rather than an internal asset.

---

> ### Author Response · Authors · 2025-11-26
> **Reply to Reviewer 8VrS Weakness 4-5**
>
> ### Weakness 4 – Comparisons with commercial models
>
> We appreciate the reviewer’s concern and agree that **not all commercial models we evaluate are dedicated SR systems**. Our intention is **not** to use them as primary SOTA baselines, but to answer two complementary questions:
>
> 1. How does LucidFlux compare to **specialized commercial SR/restoration products** that users actually deploy (e.g., **HYPIR-FLUX, Topaz, MeiTu SR**)?
> 2. How does it compare to **recent unified multimodal/generative systems** (e.g., **Gemini-NanoBanana, Seedream 4.0**) that expose restoration/upscaling interfaces?
>
> Our **main quantitative claims** are based on comparisons with **research baselines** designed for restoration (e.g., SUPIR, SeeSR, DreamClear) under matched settings. Commercial/unified systems are included **only as complementary context**, to show how a purpose-built restoration system like LucidFlux fares against tools that many practitioners use in practice. We do **not** claim Gemini or Seedream to be SOTA UIR models, and we have made this distinction explicit to avoid over-interpretation of the commercial comparisons.
>
> ---
>
> ### Weakness 5 – Inference efficiency and model size
>
> We agree that using a Flux-based backbone raises natural questions about efficiency. To address this, we benchmarked **SeeSR**, **SUPIR**, **DreamClear**, and **LucidFlux** under their official inference settings on the **same GPU** (4× upscaling to 1024², batch size 1), measuring:
>
> - captioning time (if any),
> - model inference time,
> - total time per image,
> - and parameter count.
>
> Despite using a **13.6B** Flux.1 backbone (2–4× the parameters of SeeSR/SUPIR/DreamClear), LucidFlux achieves **comparable end-to-end latency** to SeeSR and SUPIR and is **faster than DreamClear**, primarily because we remove MLLM captioning and use a lightweight conditioner:
>
> | Metric | SeeSR | SUPIR | DreamClear | LucidFlux (ours) |
> | --- | --- | --- | --- | --- |
> | Caption time (s) | 0.10 | 5.90 | 8.70 | 0.01 |
> | Inference time (s) | 22.38 | 16.60 | 28.90 | 23.61 |
> | **Total time (s)** | 22.48 | 22.50 | 37.60 | **23.61** |
> | Params (B) | 2.89 | 3.90 | 2.80 | 13.6 |
>
> These results show that, although LucidFlux scales up the backbone significantly, **its practical runtime remains in the same ballpark as prior works and is even better than DreamClear**, thanks to our caption-free SigLIP pathway and compact dual-branch conditioner. We have summarized these measurements in the revised version to make the efficiency–quality trade-off clear.
>
> [4] Scaling up to excellence: Practicing model scaling for photo-realistic image restoration in the wild. CVPR, 2024.

---

> > ### Comment · Reviewer_8VrS · 2025-11-28
> >
> > Thank you for your response. It has addressed most of my concerns, and I am willing to raise my score. However, it seems that I am unable to edit my review and score at this moment.

---

> > > ### Author Response · Authors · 2025-11-28
> > > **Thank you for your response**
> > >
> > > Dear Reviewer 8VrS,
> > >
> > > Thank you for your willingness to increase the score. Your valuable feedback has significantly improved the quality of our submission.
> > >
> > > Best regards,
> > >
> > > Authors of Paper #6814

---

### Official Review · Reviewer_arXC · 2025-10-31

**Soundness:** 3
**Presentation:** 3
**Contribution:** 3
**Rating:** 6
**Confidence:** 5

**Summary:**

This paper presents LucidFlux, a new approach to universal image restoration. The framework leverages a large-scale diffusion transformer  to restore real-world images degraded by various unknown distortions. A key contribution of this work is the development of a caption-free semantic alignment method, which replaces traditional text-based approaches with SigLIP-based semantic features derived from a lightly restored proxy image. In addition, the method incorporates a dual-branch conditioner and a timestep- and layer-adaptive modulation schedule, which together help to enhance both the structural integrity and texture details of the restored images. The framework also includes a scalable data curation pipeline to filter large scale image datasets, ensuring high-quality supervision throughout the training process.

**Strengths:**

1. LucidFlux introduces a caption-free semantic alignment method that reduces reliance on external captions. The model employs timestep- and layer-adaptive modulation to ensure efficient recovery of both fine textures and global structure. Additionally, it introduces a scalable and publicly documented data curation paradigm, ensuring high-quality supervision for large-scale image restoration.

2. The method demonstrates state-of-the-art performance across both synthetic and real-world benchmarks, consistently surpassing commercial models in both quantitative and qualitative evaluations. It offers valuable insights to the community regarding performance between commercial models and the open-source domain.

3. The paper provides a well-structured and systematic explanation of the proposed framework. The design rationale behind key components is clearly articulated, offering convincing insights into how they work in concert to enhance restoration quality.

**Weaknesses:**

1. The idea of LucidFlux is similar with DreamClear, which affects its novelty. The authors shall clarify the differences between them.

2. Some details of compared methods are missing. Clarifying these settings would improve the transparency and comparability of the results.

3. The contribution of caption-free is not so clear. Some experimental results with the same Flux.1 backbone can be useful.

**Questions:**

1. The core design of the dual-branch conditioner appears to be functionally equivalent to the paradigm used in DreamClear [1]. What is the key methodological difference that sets this work apart and establishes its novelty?

2. Although the paper mentions that experiments were conducted at a resolution of 1024, there is a lack of detail regarding the settings for comparison models, especially for super-resolution techniques. Important parameters, such as the upscale factor, are not clearly defined.

3. Although the paper claims that LucidFlux outperforms commercial models, the comparison includes Gemini-NanoBanana and Seedream 4.0, which are not specialized image restoration models. The authors do not clarify why these models were included in the comparison—whether they are considered to have performance comparable to specialized restoration models, or if there were no other commercial restoration models available for comparison. Providing more context on this choice would help clarify the relevance of these comparisons.

4. Beyond fully freezing the backbone, have you explored partial unfreezing (e.g., selected cross‑attention or upper layers)? Please report impacts on perceptual metrics and PSNR/SSIM, as well as stability and compute/memory trade‑offs.

5. Your curation uses fixed thresholds (e.g., Laplacian variance 150–8000, Sobel flat‑patch threshold ≈800 with 50% image‑level ratio, CLIP‑IQA Top‑20%). Could you report sensitivity to these choices (e.g., Top‑10/30%, different Sobel cutoffs) and their effect on metrics/semantic coverage?

6. For a clean attribution of the caption‑free design, can you add a comparison on the same Flux.1 backbone with (i) ideal GT captions and (ii) inference‑time VLM captions, and report both quality and latency?


[1] Ai Y, Zhou X, Huang H, et al. DreamClear: High-Capacity Real-World Image Restoration with Privacy-Safe Dataset Curation[J]. Advances in Neural Information Processing Systems, 2024, 37: 55443-55469.

---

> ### Author Response · Authors · 2025-11-26
> **Reply to Reviewer arXC Q1 & Q2**
>
> We sincerely thank Reviewer arXC for the thoughtful and positive assessment of our work. We are encouraged by the reviewer’s recognition of LucidFlux’s caption-free semantic alignment, dual-branch conditioning with timestep- and layer-adaptive modulation, and scalable curation pipeline, as well as the strong performance and clear presentation. Below, we clarify our methodological differences from DreamClear, specify the experimental settings (especially for super-resolution and commercial/unified baselines), and provide additional analyses on backbone freezing, curation-threshold sensitivity, and caption-based vs. caption-free variants on the same Flux.1 backbone.
>
> **Q1 & Weakness 1 – Relation to DreamClear and novelty of the dual-branch design**
>
> We thank the reviewer for pointing out the connection to DreamClear [1]. We agree that both works adopt a **dual-branch paradigm** that separates structure preservation from artifact suppression, and we explicitly acknowledge DreamClear in the paper. We do **not** claim the high-level idea of “having two branches” as our contribution. Our goal is instead to show that, once we move to a **large Flux.1 diffusion transformer**, the key novelty lies in *how* the branches are parameterized, injected, and scheduled across timesteps and layers, and how they are integrated into a fully caption-free UIR framework.
>
> Concretely, LucidFlux differs from DreamClear in three methodological aspects:
>
> 1. **Conditioner design tailored to a large Flux.1 DiT (latent-space, lightweight, and scalable).**
> DreamClear implements a heavy ControlNet-style dual branch on top of a relatively small backbone. In contrast, LucidFlux is built on a **12B Flux.1 DiT**, where simply duplicating or mirroring the backbone would be computationally prohibitive. Our dual-branch conditioner is therefore designed from the ground up to be:
>     - **Latent-feature only** (operating on VAE latents rather than full-resolution feature maps);
>     - **Shallow and non-replicated** (a few MMDiT blocks per branch, without re-instantiating the main DiT);
>
>         This makes dual-branch conditioning **practically scalable** to a large DiT, which is a different design regime from DreamClear’s smaller UNet-like backbone.
>
> 2. **Timestep- and layer-adaptive modulation as a new scheduling mechanism.**
> In LucidFlux, the dual-branch outputs are not injected in a uniform way. We introduce a **timestep- and layer-adaptive modulation schedule** that explicitly models *when* (which diffusion step) and *where* (which layer) each branch should dominate. Concretely, a small modulation head consumes normalized timestep and layer index and predicts per-channel scales and a learned gate that blends the low-quality (LQ) and lightly restored proxy (LRP) streams differently at early vs. late steps and shallow vs. deep layers. This design leverages the known temporal–hierarchical roles of DiTs (early/coarse vs. late/fine) and leads to substantial gains in our ablations. To the best of our knowledge, DreamClear does not implement such an explicit **(t, layer)-aware gating mechanism**; its dual-branch fusion is not conditioned on the joint timestep–layer context in this way.
> 3. **Integration into a caption-free, SigLIP-based semantic pathway and curated data pipeline.**
> Finally, LucidFlux’s dual-branch conditioner is tightly coupled with a **caption-free SigLIP semantic pathway** and a **documented UIR-specific data curation pipeline**. Rather than relying on inference-time captions as in DreamClear, we derive semantics from a lightly restored proxy via SigLIP and feed them through a lightweight connector into the Flux.1 text-embedding space. This removes MLLM captioning altogether and changes the role of the dual branches: one branch anchors structure and semantics derived from SigLIP on the proxy, while the other focuses on denoising and artifact removal on the LQ input under a purely image-based alignment signal. This caption-free, SigLIP-driven formulation and its associated curation pipeline are not present in DreamClear and are critical to making LucidFlux both scalable and practically deployable.
>
> **Q2 – Clarification of super-resolution settings**
>
> We thank the reviewer for pointing out this lack of detail. All super-resolution baselines in our experiments are evaluated under a **uniform 4× upscale setting**, matching the 1024×1024 resolution used for LucidFlux. Concretely, we degrade high-resolution images to low-quality inputs and ask every SR model to restore them to **4× higher spatial resolution**, after which all metrics are computed at 1024×1024. We also apply the same resizing / cropping protocol across methods to preserve aspect ratio and avoid any advantage from pre- or post-processing. We have explicitly stated the **4× upscale factor** and the shared pre-/post-processing pipeline in Sec. 4.2 of the revised manuscript to make the comparison fully transparent and reproducible.

---

> ### Author Response · Authors · 2025-11-26
> **Reply to Reviewer arXC Q3 & Q4**
>
> **Q3 – Rationale for including commercial and unified models**
>
> We appreciate the opportunity to clarify the choice of commercial baselines. Our intent is **not** to mix arbitrary problem settings, but to answer two complementary questions:
>
> (i) how LucidFlux compares to **dedicated commercial restoration/SR products** that users rely on in practice; and
>
> (ii) how it compares to **recent unified generative models** that also offer image restoration capabilities.
>
> Most of our commercial baselines—**HYPIR-FLUX, Topaz, and MeiTu SR**—are **specialized super-resolution or enhancement tools**, and we treat them as practical reference points for real-world usage. In contrast, **Gemini-NanoBanana** and **Seedream 4.0** are **general multimodal or generative systems** that nonetheless expose restoration/upscaling interfaces and can handle the same degraded inputs (as illustrated in Fig. 8). We include them **not as dedicated UIR baselines**, but to provide a broader view of how strong unified models perform on restoration compared to a purpose-built UIR system.
>
> In the revision, we have (i) explicitly labeled which baselines are specialized SR/restoration products versus unified generative models, and (ii) clarified that we do **not** claim Gemini-NanoBanana or Seedream 4.0 to be state-of-the-art UIR systems, but rather use them to contextualize LucidFlux against the current generation of widely deployed commercial models.
>
> **Q4: Impact of Partial Backbone Unfreezing**
>
> We thank the reviewer for the suggestion. We have explored different backbone update strategies on the **RealLQ250** dataset, including **attention-only updates**, **full fine-tuning of all parameters**, and our proposed **frozen-backbone adaptation**. The detailed comparison is provided in **Supplementary Material A.4**.
>
> As summarized in the table below, updating only the attention layers leads to **stable training** but hits a **performance ceiling** (CLIP-IQA 0.654 / MUSIQ 66.27), indicating that adapting cross-attention alone is insufficient for handling complex real-world degradations. Full fine-tuning improves the model’s adaptability but can cause **catastrophic forgetting**, reducing perceptual quality (CLIP-IQA 0.594 / MUSIQ 63.30).
>
> Our **frozen-backbone adaptation** achieves the **best perceptual performance** (CLIP-IQA 0.7122 / MUSIQ 73.01) while maintaining stable optimization. Although its **memory footprint** is similar to full fine-tuning (76.53 GB vs. 76.16 GB), it **avoids destabilizing the generative prior**, offering a favorable trade-off between perceptual performance, stability, and computational cost.
>
> | Training Strategy | CLIP-IQA ↑ | NIQE ↓ | MUSIQ ↑ | Memory |
> | --- | --- | --- | --- | --- |
> | Attention-only Fine-Tuning | 0.654 | 3.52 | 66.27 | 60.30 GB |
> | Full Fine-Tuning | 0.594 | 3.796 | 63.30 | 76.16 GB |
> | Ours | 0.7122 | 3.6742 | 73.01 | 76.53 GB |
>
> Overall, our approach provides the **best balance between perceptual quality, training stability, and resource usage. These results suggest that for large DiTs, adaptation is best absorbed by lightweight conditioners while keeping the generative prior frozen.**

---

> > ### Author Response · Authors · 2025-11-26
> > **Reply to Reviewer arXC Q5**
> >
> > **Q5 – Sensitivity to data curation thresholds**
> >
> > We thank the reviewer for this insightful question. We have conducted a systematic sensitivity study of our curation thresholds (blur/Laplacian, flat-patch Sobel cutoff and ratio, CLIP-IQA top-k) on the same RealLQ250 pool, and report the results in Appendix A.6.
> >
> > 1. **Effect on dataset size and semantic coverage.**
> >
> >     Starting from the raw pool (≈2.9M images), our baseline thresholds (Laplacian 150–8000, flat-patch Sobel variance 800 with 50% image-level ratio, CLIP-IQA Top-20%) retain ≈234k samples. Varying each component individually (e.g., tightening blur to 200–6000, relaxing flatness to Sobel 600 & 60%, or changing CLIP-IQA to Top-10%/30%) moves the retained set between ≈123k and ≈330k images , as summarized in the table below. To assess semantic coverage, we sample 1k/5k/10k images from each configuration and visualize SigLIP embeddings via t-SNE, as shown in Figure 7. Under a **fixed sample size** (e.g., 10k), the semantic distribution remains very similar across different threshold settings, whereas increasing the sample size under any configuration expands coverage. This indicates that **semantic diversity is primarily driven by dataset scale, not by small perturbations of individual thresholds.**
> >
> >     | Setting | Threshold | Sample Number |
> >     | --- | --- | --- |
> >     | Pool | None | 2,900,747 |
> >     | Baseline | Blur 150–8000, Flat 800, 50%, CLIP-IQA 20% | 234,287 |
> >     | Blur Var. Set 1 | Blur 200–6000, Flat 800, 50%, CLIP-IQA 20% | 219,966 |
> >     | Blur Var. Set 2 | Blur 100–10000, Flat 800, 50%, CLIP-IQA 20% | 245,090 |
> >     | Flat Var. Set 3 | Blur 150–8000, Flat 1000, 40%, CLIP-IQA 20% | 163,482 |
> >     | Flat Var. Set 4 | Blur 150–8000, Flat 600, 60%, CLIP-IQA 20% | 301,637 |
> >     | CLIP-IQA Var. Set 5 | Blur 150–8000, Flat 800, 50%, CLIP-IQA 10% | 123,007 |
> >     | CLIP-IQA Var. Set 6 | Blur 150–8000, Flat 800, 50%, CLIP-IQA 30% | 330,618 |
> > 2. **Effect on image quality metrics.**
> >
> >     We further evaluate curated subsets from each configuration using seven no-reference IQA metrics (CLIP-IQA+, QAlign, MUSIQ, MANIQA, NIMA, CLIP-IQA, NIQE), as listed in the table below. Our baseline configuration consistently achieves **the best or near-best scores across metrics**. Stricter thresholds (e.g., CLIP-IQA Top-10%) slightly reduce diversity and yield only marginal gains or even small drops in some metrics, while more relaxed thresholds (e.g., Top-30% or looser flatness constraints) admit more low-quality images and moderately degrade IQA scores. Overall, the differences across reasonable threshold ranges are modest, confirming that our curation pipeline is **robust**, and that the chosen baseline offers a **well-balanced trade-off between scale, semantic coverage, and image quality**.
> >
> >     | Setting | CLIP-IQA+ ↑ | QAlign ↑ | MUSIQ ↑ | MANIQA ↑ | NIMA ↑ | CLIP-IQA ↑ | NIQE ↓ |
> >     | --- | --- | --- | --- | --- | --- | --- | --- |
> >     | Baseline | **0.7406** | **4.3935** | **73.0088** | **0.5589** | 5.4836 | **0.7122** | 3.6742 |
> >     | Blur Var. Set 1 | 0.7023 | 4.3041 | 71.3748 | 0.5057 | 5.4614 | 0.6622 | 3.4348 |
> >     | Blur Var. Set 2 | 0.7072 | 4.2464 | 71.8887 | 0.5140 | **5.5129** | 0.6590 | 3.8099 |
> >     | Flat Var. Set 3 | 0.7011 | 4.2164 | 71.3910 | 0.5161 | 5.4096 | 0.6573 | 3.6507 |
> >     | Flat Var. Set 4 | 0.6913 | 4.2795 | 70.2861 | 0.4973 | 5.4570 | 0.6584 | **3.3471** |
> >     | CLIP-IQA Var. Set 5 | 0.6749 | 3.9777 | 69.8807 | 0.4827 | 5.2303 | 0.6226 | 3.5555 |
> >     | CLIP-IQA Var. Set 6 | 0.6897 | 4.2487 | 70.3159 | 0.4891 | 5.3737 | 0.6418 | 3.4848 |
> >
> > We have added these quantitative results and visualizations to Appendices A6.1 and A6.2 in the revised manuscript. In the main text, we briefly summarize the key takeaway: LucidFlux's performance is **not brittle** to the specific numeric choices of the thresholds, and the main driver is having a sufficiently large, high-quality curated set rather than fine-tuning any single cutoff.

---

> > > ### Author Response · Authors · 2025-11-26
> > > **Reply to Reviewer arXC Q6**
> > >
> > > **Q6 & Weakness 3 – Caption-free design vs. GT / VLM captions**
> > >
> > > We thank the reviewer for this helpful suggestion. To cleanly attribute the effect of the caption-free design, we conducted an ablation on the **same Flux.1 backbone** with three variants:
> > >
> > > 1. **GT captions** – captions generated by LLaVA from the *ground-truth clean* images (oracle setting, not available in practice);
> > > 2. **VLM captions** – captions generated by LLaVA from the *lightly restored proxy* as DreamClear used at inference time;
> > > 3. **Caption-free (ours)** – no captions, using our **SigLIP-based semantic pathway** from the lightly restored proxy.
> > >
> > > All variants share the same architecture (dual-branch conditioner + TLCM) and training setup; only the semantic source differs.
> > >
> > > On **RealLQ250**, we observe that the caption-free variant achieves the **best overall perceptual quality** while also eliminating captioning latency, as shown in the table below.:
> > >
> > > - Our caption-free model attains the highest **MUSIQ** (70.20) and **NIMA** (5.11) and the lowest **NIQE** (4.29), slightly outperforming the GT-caption oracle (MUSIQ 70.17, NIMA 5.06, NIQE 4.68) and the VLM-caption variant (MUSIQ 69.44, NIMA 5.05, NIQE 4.62).
> > > - CLIP-IQA+ and QAlign remain comparable across all three variants, confirming that we do not sacrifice semantic alignment.
> > > - Crucially, both caption-based variants incur roughly **+10 s additional latency per image** due to LLaVA inference.
> > >
> > > | Method | CLIP-IQA+ ↑ | Q-Align ↑ | MUSIQ ↑ | MANIQA ↑ | NIMA ↑ | CLIP-IQA ↑ | NIQE ↓ | Latency |
> > > | --- | --- | --- | --- | --- | --- | --- | --- | --- |
> > > | GT caption (LLaVA on GT) | 0.7111 | 3.8713 | 70.1654 | 0.5686 | 5.0632 | 0.7032 | 4.6775 | +10.426 s |
> > > | VLM caption (LLaVA on LRP) | 0.7060 | 3.8168 | 69.4371 | 0.5533 | 5.0475 | 0.7030 | 4.6249 | +10.057 s |
> > > | Ours (caption-free) | 0.7074 | 3.7555 | **70.2005** | 0.5437 | **5.1072** | 0.6783 | **4.2893** | +0.012 |
> > >
> > > We have report these results in the revised manuscript (Table 6 / Appendix A.3) and reference them in the main text. Together with our analysis of VLM caption biases (Appendix A.1–A.2), this ablation shows that LucidFlux’s gains come from the **caption-free SigLIP semantic alignment plus dual-branch/TLCM design**, rather than from “hidden” or stronger textual supervision, while also making the method more practical for real-world deployment.

---

### Official Review · Reviewer_V2EC · 2025-10-31

**Soundness:** 3
**Presentation:** 3
**Contribution:** 3
**Rating:** 8
**Confidence:** 4

**Summary:**

The authors propose LucidFlux, a Flux.1 diffusion transformer adapted to the universal image restoration task.

Starting from a frozen Flux.1 backbone, four trainable modules are added which are trained on 1.36 million LQ-HQ image pairs. These image pairs are synthesized using the Real-ESRGAN degradation pipeline, starting from 342k HQ images which the authors in turn obtain by applying a proposed filtering pipeline to 2.9 million candidate images.

The proposed LucidFlux model obtains impressive qualitative results and performs well compared to recent open- and closed-source methods.

**Strengths:**

- The paper is well written overall, components of the proposed approach are explained well in Section 3. I found the paper quite interesting and enjoyed reading it overall.
- LucidFlux seems to perform well compared to recent open- and closed-source methods.
- LucidFlux achieves seemingly very impressive qualitative/visual results.

**Weaknesses:**

- Incorrect citation formatting across basically the entire paper. For example, line 69: "Discriminative restorers based on CNNs and Transformers Dong et al. (2016); Liang et al. (2021); Zamir et al. (2022) perform well on..." --> "Discriminative restorers based on CNNs and Transformers (Dong et al., 2016; Liang et al., 2021; Zamir et al., 2022) perform well on...". This makes parts of the paper quite difficult to read, but it's a simple fix.
- Some details regarding the data collection and model training are missing.

**Questions:**

- In Section 3.4 you write "First, we collect 2.3M images from the Internet". This is quite vague indeed? Could you give more details on how and from which sources you collect these images?
- Perhaps I'm missing something obvious, but what loss is used to train the model?




Minor things:
- Duplicate reference of "Eirikur Agustsson and Radu Timofte. Ntire 2017 challenge on single image super-resolution: Dataset and study"?
- "MLLM" should perhaps be defined when it's used in the abstract and at the beginning of Section 3.
- In Figure 2, $\phi_{LQ}$ and $\phi_{LRP}$ could perhaps be added as output of the two Dual-Branch Conditioners blocks?
- Line 218: LQ and LRP might not need to be defined again here?
- Line 223: "These parameters effect" --> These parameters affect"?
- In equation 4, I would probably consider changing the circle symbol to something else.
- "T2I" is defined at the start of Section 3.4, but not at the start of 3.3.
- The "Lvmin Zhang, Anyi Rao, and Maneesh Agrawala. Adding conditional control to text-to-image diffusion models" reference is missing year and venue.
- "IQA" should probably be defined somewhere.
- Line 453: "and three scores are enlarged after adding caption-free" --> "and all three scores are improved after adding caption-free"?
- Line 464: "and the identical IQA metrics are used in the open-source comparisons" --> "and the identical IQA metrics are used as in the open-source comparisons"?
- Line 482, "by large DiTs.Across real": typo.

---

> ### Author Response · Authors · 2025-11-26
> **Reply to Reviewer V2EC**
>
> We sincerely thank Reviewer V2EC for the thoughtful and positive assessment of our work. We are encouraged by the reviewer’s comments that the paper is well written, that the components of LucidFlux are clearly explained, and that the model delivers strong quantitative performance and very impressive visual results compared to recent open- and closed-source methods. Below, we address the remaining questions on data collection and training details, and we summarize the textual and formatting revisions we have made in response to the reviewer’s suggestions.
>
> ### Q1 – Data collection details
>
> We thank the reviewer for asking for more detail. The 2.3M images are collected **exclusively from copyright-friendly sources**, namely **Pexels** and **Unsplash**, via their **official APIs**. We use the **23,914 unique keywords extracted from** the Unsplash Lite dataset to query both platforms, which gives broad coverage across categories (people, cities, landscapes, products, etc.). These 2.3M internet images are then **combined with 557K photos from Photo-Concept-Bucket**, yielding the **2.9M candidate pool** described in Sec. 3.4. During crawling we discard near-duplicate and extremely low-resolution images to avoid trivial or unusable samples. All images are obtained under the standard Pexels/Unsplash licenses, which permit research use and derivative works. In the revision, we have (i) stated these sources explicitly, (ii) described the keyword-based crawling procedure to make the candidate pool **fully reproducible**.
>
> ---
>
> ### Q2 – Training loss and missing training details
>
> We apologize for not making the training objective explicit. LucidFlux is trained with an **L2 reconstruction loss (mean squared error)** in the **VAE latent space**, following the standard **flow-matching** formulation: we minimize the MSE between the model’s predicted velocity field and the corresponding flow-matching target in latent space. This is the loss implied by our flow-matching equation but was not clearly spelled out in the text. In the revised manuscript, we have explicitly named this loss in Sec. 4.1 where we describe the training setup and link it to the flow-matching objective. In addition, to address the broader concern that some training details are missing, we have expanded Sec. 4.1 to specify the optimizer, learning-rate schedule, batch size, and the total number of training steps/epochs.
>
> ---
>
> ### Response to minor issues & Weakness 1 (citation formatting)
>
> We sincerely thank the reviewer for the careful reading and detailed suggestions. We have **systematically fixed the citation formatting throughout the paper**, e.g., using “(Dong et al., 2016; Liang et al., 2021; Zamir et al., 2022)” instead of inline author lists, and performed a full proofreading pass to unify citation style, acronyms, and notation. We also addressed all specific minor issues mentioned: removed the duplicate NTIRE reference; defined **MLLM**, **IQA**, and **T2I** at their first use; corrected typos (e.g., “effect” → “affect”, “three scores are enlarged” → “three scores are improved”); adjusted Figure 2 to better indicate the outputs of the dual-branch conditioners; changed the symbol in Eq. (4) to avoid ambiguity; completed missing reference details (e.g., for “Adding conditional control to text-to-image diffusion models”); and fixed small formatting glitches (e.g., missing spaces). These changes materially improve the **readability and consistency** of the manuscript.

---

### Official Review · Reviewer_aM2Z · 2025-11-02

**Soundness:** 4
**Presentation:** 4
**Contribution:** 2
**Rating:** 4
**Confidence:** 5

**Summary:**

This paper proposes LucidFlux, a caption-free UIR framework that adapts the large diffusion transformer Flux.1 without relying on image captions.

**Strengths:**

1. The paper is well-written, the results are great,  and the experiments are solid.
2. A lightweight dual-branch conditioner is incorporated that injects signals from both the degraded input and a lightly restored proxy.
3. A novel timestep- and layer-adaptive modulation schedule then routes these conditioning cues throughout the backbone's hierarchy.
4. To ensure semantic alignment without the latency and instability of text prompts or MLLM captions, caption-free alignment is enforced using SigLIP features extracted from the proxy image.

**Weaknesses:**

1. It is claimed that "Universal image restoration (UIR) aims to recover images degraded by unknown mixtures while preserving semantics—conditions under which discriminative restorers and UNet-based diffusion priors often oversmooth, hallucinate, or drift." However, there are lots of DiT-based diffusion priors in UIR. This main challenge is not solid.
2. As for the first contribution, "a lightweight dual-branch conditioner", it seems very similar to DiffBIR. The main difference is that DIffBIR only uses a lightly restored proxy as conditions.
3. As for the second contribution, "timestep- and layer-adaptive modulation", it is also a common technique for injecting conditions.
4. "A SigLIP-based module preserves semantic consistency without prompts or captions." The motivation is great but the contribution is minor.

**Questions:**

1. The paper positions itself by claiming that UNet-based diffusion priors in UIR often lead to oversmoothing, hallucination, or semantic drift. However, numerous recent and successful UIR methods utilize DiT-based architectures. Could the authors clarify the specific limitations of these existing DiT-based priors to more solidly justify the necessity of their proposed approach and distinguish it from this relevant body of work?
2. The proposed dual-branch conditioner appears functionally similar to the conditioning mechanism in DiffBIR, which also utilizes a restored image proxy. Could the authors elaborate in detail on the fundamental technical and conceptual differences between their conditioner and prior arts, clarifying the specific novelty and advantages introduced by their dual-branch design?
3. The timestep- and layer-adaptive modulation is presented as a key contribution. Given that adaptive feature modulation is a well-established technique for injecting conditions in diffusion and other generative models, what is the specific innovation in your implementation? Please clarify how your method differs from and advances upon this common paradigm.
4. Why not compare DiffBIR across all experiments?

Minor Suggestion:
1. It is recommended to use \citep rather than \cite.

---

> ### Author Response · Authors · 2025-11-26
> **Reply to Reviewer aM2Z Q1**
>
> We sincerely thank Reviewer aM2Z for the detailed and constructive review. We appreciate the positive assessment of the paper’s soundness, presentation quality, and empirical results, as well as the clear summary of our key components. In what follows, we clarify (i) how LucidFlux is positioned relative to existing DiT-based UIR methods, (ii) the technical differences between our lightweight dual-branch conditioner and DiffBIR-style designs, (iii) the specific role and novelty of our timestep- and layer-adaptive modulation and caption-free SigLIP semantic module, and (iv) the updated experiments that now include DiffBIR across all benchmarks, along with minor formatting and citation revisions.
>
> Q1:**Limitations of Existing DiT-Based UIR Methods and Positioning of LucidFlux**
>
> We thank the reviewer for highlighting that recent UIR works already employ DiT-based diffusion priors. We agree that our original introduction placed too much emphasis on UNet-based priors and did not sufficiently discuss this line of work. The goal of LucidFlux is therefore not to argue that “UIR is still dominated by UNet-based priors,” but to address a *complementary* question: **how to adapt a large, generic text-to-image DiT such as Flux.1 to real-world UIR under mixed degradations and without captions.**
>
> Concretely, we see current DiT-based UIR methods as having two main limitations that LucidFlux is designed to tackle:
>
> - **Backbone scale and capacity.**
>
>     Existing DiT-based UIR models typically adopt relatively small backbones (e.g., ≈0.6–1B parameters, such as PixArt-α in DreamClear), trained specifically for restoration. Under severe mixed real-world degradations, such models struggle to simultaneously maintain global structural coherence and recover fine textures. In contrast, LucidFlux adapts a *frozen 12B Flux.1 backbone* with only a 1.6B adapter. As shown in Table 3, despite this much larger backbone, our caption-free design achieves end-to-end runtime comparable to SeeSR, SUPIR, and DreamClear (which use 0.6–3.5B backbones plus captioning), while delivering stronger IQA and semantic fidelity on RealLQ250 and DRealSR. This suggests that our contribution lies in *how a large DiT prior is adapted* to UIR, rather than merely switching from UNet to DiT.
>
> - **Conditioning formulation and caption dependence.**
>
>     Many DiT-based UIR systems interface with the backbone via heavy ControlNet-like adapters driven by captions. This (i) duplicates a substantial portion of the DiT, inflating parameters and latency; (ii) injects conditions in a layer-aligned but largely *timestep-agnostic* manner, without explicitly exploiting the temporal–hierarchical structure of the diffusion process; and (iii) reintroduces caption latency and caption bias (e.g., degradations being literally described in the caption). LucidFlux instead uses a lightweight Dual-Branch Conditioner (DBC) plus timestep- and layer-adaptive modulation (TLCM) to condition the frozen Flux.1 backbone, and a SigLIP-based semantic alignment module that is entirely caption-free at both training and inference. Our ablations (Table 4) show that starting from a DBC-only model trained on LSDIR (0.585/0.609/61.582 CLIP-IQA/CLIP-IQA+/MUSIQ), progressively adding SigLIP alignment, TLCM, and our curated high-quality data improves performance to 0.7122/0.7406/73.0088, indicating that this specific adaptation scheme is crucial to fully exploiting the large DiT prior.
>
>
> In summary, we clarify}that LucidFlux does *not* claim DiT-based priors have not been explored for UIR. Instead, our focus is on demonstrating that a **caption-free, large-scale Flux.1 prior, combined with a carefully designed conditioning and data pipeline, yields a different and practically important operating point** compared to existing small-DiT, ControlNet-style, caption-dependent restorers.
>
> **We have slightly revised Section 1 (Introduction) and Section 2 (Related Work) to explicitly discuss recent DiT-based UIR methods and to clarify that our focus is on adapting a large Flux.1 prior with caption-free conditioning, rather than on claiming that UIR is dominated by UNet-based priors.**

---

> > ### Author Response · Authors · 2025-11-26
> > **Reply to Reviewer aM2Z Q2**
> >
> > Q2: **Fundamental Differences Between Our Dual-Branch Conditioner and DiffBIR**
> >
> > We appreciate the reviewer’s concern that our dual-branch conditioner may appear similar to DiffBIR. While both methods make use of a restored proxy, **their architecture, conditioning mechanism, and use of inputs are fundamentally different**. Our Dual-Branch Conditioner (DBC) is designed as a **lightweight adapter for a frozen 12B Flux.1**, whereas DiffBIR follows a **heavy, ControlNet-style copy-and-add** paradigm.
> >
> > **(1) Heavyweight copy vs. lightweight latent adapter**
> >
> > DiffBIR adopts a ControlNet-like design: it **copies substantial encoder/backbone capacity** to process the conditional input and injects a **layer-aligned feature tower** into the main network, so the conditional branch scales with backbone depth. In contrast, DBC operates at **VAE latent resolution** and consists of only **two MMDiT blocks per branch**, with depth independent of the 12B Flux.1 backbone, which remains frozen. This keeps the adapter small in parameters/FLOPs and makes adapting a large generic DiT to IR practical, rather than duplicating backbone stages.
> >
> > **(2) Feature addition vs. per-(t,l) modulation**
> >
> > In DiffBIR, the copied branch outputs per-layer feature maps that are **added/concatenated** into backbone activations, potentially overriding them and not explicitly encoding how conditioning strength should evolve across timesteps and layers. In LucidFlux, DBC produces two latent-level streams ( $\phi_{\text{LQ}}$ ) and ( $\phi_{\text{LRP}}$ ); the TLCM module then turns these into **per-(t,l) modulation parameters** applied in normalization layers of Flux.1. Thus we **modulate** (scale/shift) the backbone’s hidden states rather than replacing them, and the influence of the condition is explicitly tied to both timestep and depth, preserving the backbone’s diffusion dynamics.
> >
> > **(3) Single proxy vs. dual complementary streams with a learned gate**
> >
> > DiffBIR conditions on a **single proxy stream**, which can inherit artifacts or oversmoothing from the restored image. Our conditioner is **explicitly dual-branch**: the LQ branch is anchored to the degraded input (richer local detail), and the LRP branch to the lightly restored proxy (more stable global structure and semantics). TLCM learns a gate ( $\gamma_{t,l}$ ) over timesteps and layers to **interpolate between LQ and LRP**. Empirically, early timesteps/deep layers lean more on LRP (structure-first), while later timesteps/shallow layers gradually increase the contribution of LQ (detail refinement). This **coarse-to-fine, data-driven fusion of LQ and LRP** is not present in DiffBIR’s single-stream conditioning.
> >
> > In short, LucidFlux is **inspired by dual-branch conditioning** but differs from DiffBIR by (i) using a shallow latent-level adapter instead of a copied backbone, (ii) steering a frozen 12B Flux.1 via per-(t,l) normalization-based modulation rather than direct feature addition, and (iii) exploiting complementary LQ/LRP inputs through a learned temporal–hierarchical gate instead of a single proxy.

---

> ### Author Response · Authors · 2025-11-26
> **Reply to Reviewer aM2Z Q3**
>
> **Q3: Specific Innovation in Our Timestep- and Layer-Adaptive Modulation (TLCM)**
>
> We agree that FiLM/AdaLN-style adaptive modulation is a standard primitive. Our contribution is **not** a new low-level operation, but **how** this primitive is structured and used in the large-DiT, caption-free UIR setting with a dual-branch conditioner.
>
> **(1) Joint timestep–layer scheduling instead of a single axis.**
>
> In many diffusion systems, conditioning is either applied **uniformly across layers** given a global condition, or made **timestep-dependent only** (e.g., time-dependent AdaLN). TLCM instead takes normalized timestep (t/T) and layer index (l/L) together with ($\phi_{\text{LQ}}$, $\phi_{\text{LRP}}$), and predicts **per-(t,l) modulation parameters**, yielding a learned 2D schedule that aligns with the DiT hierarchy: early timesteps/deeper layers bias toward coarse structure, later timesteps/shallower layers toward detail refinement.
>
> **(2) Dual-branch gate over LQ and LRP, rather than a single condition stream.**
>
> TLCM is specifically designed for our dual-branch conditioner. For each ((t,l)), it outputs both modulation parameters and a gate $\gamma_{t,l}$ that **interpolates between** $\phi_{\texttt{LQ}}$ and $\phi_{\texttt{LRP}}$. Empirically, early timesteps and deep layers place more weight on the proxy branch (LRP) to stabilize global structure and semantics under heavy degradations, while later timesteps and shallow layers progressively increase the contribution of the LQ branch to recover fine textures. This **coarse-to-fine, data-driven trade-off between LQ and LRP across the diffusion trajectory** is not modeled in prior UIR modulation schemes.
>
> **(3) Quantitative impact in the large-DiT, caption-free regime.**
>
> The effect of TLCM is measurable: on RealLQ250 (Table 4), starting from a model with DBC + SigLIP alignment, **adding TLCM alone** improves CLIP-IQA / CLIP-IQA+ / MUSIQ from **0.600 / 0.620 / 62.000** to **0.622 / 0.635 / 65.500**. Together with caption-free semantics and curated high-quality data, this leads to our final **0.7122 / 0.7406 / 73.0088** scores. This indicates that explicitly coupling modulation to both timestep and depth, and using it to orchestrate the LQ/LRP balance, goes beyond a simpler timestep-only or layer-uniform modulation.
>
> In summary, TLCM advances the *use* of adaptive modulation in UIR by treating ((t,l)) as a joint modulation domain and binding this 2D schedule to a dual-branch gate between degraded and proxy features, which is what allows LucidFlux to steer a frozen 12B Flux.1 prior in a degradation-aware manner without duplicating the backbone or relying on captions.

---

> ### Author Response · Authors · 2025-11-26
> **Reply to Reviewer aM2Z Q4**
>
> **Q4:Updated Comparisons: DiffBIR Added Across Experiments**
>
> We sincerely thank the reviewer for the suggestion. We have now included DiffBIR in experiments for a more comprehensive comparison. Please see the table below for the updated results. We have also updated Table 1 in the manuscript accordingly. This additional comparison makes our experiments more thorough and supports our analysis more convincingly.
>
> | Benchmark | Dataset | Metric | ResShift | StableSR | SinSR | DiffBIR | SeeSR | DreamClear | SUPIR | LucidFlux (Ours) |
> | --- | --- | --- | --- | --- | --- | --- | --- | --- | --- | --- |
> | Caption-Free |  |  | ✓ | ✓ | ✓ | ✓ | ✗ | ✗ | ✗ | ✓ |
> | Real-world | DRealSR | CLIP-IQA+ ↑ | 0.4655 | 0.3732 | 0.5402 | 0.6475 | 0.6257 | 0.4461 | 0.5494 | **0.6748** |
> |  |  | Q-Align ↑ | 2.6311 | 2.1245 | 3.1334 | 3.0490 | 3.2745 | 2.4213 | 3.4720 | **3.6919** |
> |  |  | MUSIQ ↑ | 40.9795 | 29.6691 | 53.9138 | 60.0759 | 61.3222 | 35.1911 | 54.9279 | **66.6833** |
> |  |  | MANIQA ↑ | 0.2687 | 0.2402 | 0.3455 | 0.4900 | 0.4505 | 0.2675 | 0.3482 | **0.4985** |
> |  |  | NIMA ↑ | 4.3178 | 3.9048 | 4.6226 | 4.6543 | 4.6401 | 3.9368 | 4.5063 | **4.9625** |
> |  |  | CLIP-IQA ↑ | 0.4964 | 0.3383 | 0.6631 | 0.6781 | 0.6760 | 0.4360 | 0.5309 | **0.6879** |
> |  |  | NIQE ↓ | 10.3005 | 8.6022 | 6.9800 | 6.4852 | 6.4502 | 7.0163 | 5.9091 | **4.7034** |
> |  |  |  |  |  |  |  |  |  |  |  |
> |  | RealSR | CLIP-IQA+ ↑ | 0.5005 | 0.4408 | 0.5416 | 0.6543 | 0.6731 | 0.5331 | 0.5640 | **0.7074** |
> |  |  | Q-Align ↑ | 3.1045 | 2.5087 | 3.3615 | 3.3157 | 3.6073 | 3.0044 | 3.4682 | **3.7555** |
> |  |  | MUSIQ ↑ | 49.50 | 39.98 | 57.95 | 61.7750 | 67.57 | 49.48 | 55.68 | **70.20** |
> |  |  | MANIQA ↑ | 0.2976 | 0.2356 | 0.3753 | 0.4744 | 0.5087 | 0.3092 | 0.3426 | **0.5437** |
> |  |  | NIMA ↑ | 4.7026 | 4.3639 | 4.8282 | 4.8192 | 4.8957 | 4.4948 | 4.6401 | **5.1072** |
> |  |  | CLIP-IQA ↑ | 0.5283 | 0.3521 | 0.6601 | 0.6805 | **0.6993** | 0.5390 | 0.4857 | 0.6783 |
> |  |  | NIQE ↓ | 9.0674 | 6.8733 | 6.4682 | 6.0700 | 5.4594 | 5.2873 | 5.2819 | **4.2893** |
> |  |  |  |  |  |  |  |  |  |  |  |
> |  | RealLQ250 | CLIP-IQA+ ↑ | 0.5529 | 0.5804 | 0.6054 | 0.6918 | 0.7034 | 0.6810 | 0.6532 | **0.7406** |
> |  |  | Q-Align ↑ | 3.6318 | 3.5586 | 3.7451 | 3.9757 | 4.1423 | 4.0640 | 4.1347 | **4.3935** |
> |  |  | MUSIQ ↑ | 59.50 | 57.25 | 65.45 | 67.5313 | 70.38 | 67.08 | 65.81 | **73.01** |
> |  |  | MANIQA ↑ | 0.3397 | 0.2937 | 0.4230 | 0.4899 | 0.4895 | 0.4400 | 0.3826 | **0.5589** |
> |  |  | NIMA ↑ | 5.0624 | 5.0538 | 5.2397 | 5.3132 | 5.3146 | 5.2200 | 5.0806 | **5.4836** |
> |  |  | CLIP-IQA ↑ | 0.6129 | 0.5160 | **0.7166** | 0.7137 | 0.7063 | 0.6950 | 0.5767 | 0.7122 |
> |  |  | NIQE ↓ | 6.6326 | 4.6236 | 5.4425 | 5.1193 | 4.4383 | 3.8700 | **3.6591** | 3.6742 |
> |  |  |  |  |  |  |  |  |  |  |  |
> | Synthetic | DIV2K-Val | CLIP-IQA+ ↑ | 0.5583 | 0.5760 | 0.6128 | 0.6973 | 0.7116 | 0.6585 | 0.6719 | **0.7492** |
> |  |  | Q-Align ↑ | 3.5761 | 3.4226 | 3.7336 | 3.8509 | 4.1167 | 3.9323 | 4.1659 | **4.5311** |
> |  |  | MUSIQ ↑ | 60.5932 | 57.4246 | 66.0906 | 69.1822 | 71.4947 | 65.8187 | 67.9074 | **73.9045** |
> |  |  | MANIQA ↑ | 0.3421 | 0.2902 | 0.4341 | 0.5015 | 0.5104 | 0.4369 | 0.4148 | **0.5819** |
> |  |  | NIMA ↑ | 5.0430 | 5.0341 | 5.1810 | 5.1941 | 5.2709 | 5.1663 | 5.1516 | **5.4884** |
> |  |  | CLIP-IQA ↑ | 0.6017 | 0.5002 | **0.7166** | 0.7143 | 0.7149 | 0.6663 | 0.5848 | 0.7034 |
> |  |  | NIQE ↓ | 6.1976 | 4.9810 | 5.3679 | 4.8437 | 4.2823 | 4.1634 | 3.7701 | **3.7283** |
> |  |  | PSNR ↑ | 18.3802 | 18.3269 | 18.0956 | **20.0389** | 18.2529 | 17.5701 | 17.7567 | 15.4393 |
> |  |  | SSIM ↑ | 0.4394 | **0.4819** | 0.4259 | 0.5242 | 0.4684 | 0.4291 | 0.4482 | 0.3837 |
> |  |  | LPIPS ↓ | 0.3738 | 0.3933 | 0.3919 | 0.3582 | **0.3497** | 0.3621 | 0.3785 | 0.4312 |
> |  |  |  |  |  |  |  |  |  |  |  |
> |  | LSDIR-Val | CLIP-IQA+ ↑ | 0.5248 | 0.5576 | 0.5582 | 0.6977 | 0.7258 | 0.6995 | 0.7126 | **0.7440** |
> |  |  | Q-Align ↑ | 3.5317 | 3.4878 | 3.7095 | 3.9514 | 4.2997 | 4.2391 | 4.3468 | **4.5959** |
> |  |  | MUSIQ ↑ | 57.6691 | 57.0838 | 63.9586 | 68.4680 | 72.0142 | 70.7186 | 70.3340 | **74.1923** |
> |  |  | MANIQA ↑ | 0.3408 | 0.2990 | 0.4131 | 0.5356 | 0.5529 | 0.5059 | 0.4482 | **0.5979** |
> |  |  | NIMA ↑ | 5.0916 | 5.0628 | 5.3353 | 5.3566 | 5.4245 | 5.3773 | 5.3692 | **5.6221** |
> |  |  | CLIP-IQA ↑ | 0.5691 | 0.4991 | **0.6766** | 0.7066 | 0.7314 | 0.6941 | 0.6105 | 0.6836 |
> |  |  | NIQE ↓ | 6.4447 | 4.2104 | 5.1771 | 4.4015 | 3.9402 | 3.3318 | **2.9610** | 3.5571 |
> |  |  | PSNR ↑ | **17.3040** | 17.1480 | 16.8241 | 18.8862 | 17.0782 | 16.2114 | 16.1598 | 14.8688 |
> |  |  | SSIM ↑ | 0.3935 | 0.4026 | 0.3710 | 0.4652 | **0.4113** | 0.3823 | 0.3636 | 0.3697 |
> |  |  | LPIPS ↓ | 0.4824 | 0.4655 | 0.4637 | 0.4344 | 0.3969 | **0.3720** | 0.4408 | 0.4148 |

---

> > ### Author Response · Authors · 2025-11-26
> > **Reply to Reviewer aM2Z Weakness 4**
> >
> > Response to Weakness 4: **Clarifying the Technical and Practical Contribution of the Caption-Free SigLIP Semantic Module**
> >
> > We appreciate the reviewer’s comment. We agree that simply “using SigLIP” as another encoder would be a minor change; our contribution lies in **how SigLIP is integrated to make a large-DiT UIR system truly caption-free and stable**:
> >
> > - **Connector into the Flux.1 text space.**
> >
> >     SigLIP features are *not* used as raw global embeddings. We pass them through a lightweight Connector that projects them into the **text-conditioning space expected by Flux.1**, so that they form a compatible semantic context for the frozen 12B backbone rather than an extra visual branch.
> >
> > - **Semantics from the proxy, not from degraded inputs.**
> >
> >     We extract SigLIP features from the **lightly restored proxy (LRP)** instead of directly from the degraded image, which substantially reduces degradation-induced semantic drift and stabilizes the guidance under heavy real-world degradations.
> >
> > - **Enabling a fully caption-free UIR pipeline with measurable gains.**
> >
> >     Combined with the Dual-Branch Conditioner and TLCM, this design allows LucidFlux to be **caption-free at both training and inference**. Table 4 shows that, starting from a DBC-only model trained on LSDIR (0.585 / 0.609 / 61.582 CLIP-IQA / CLIP-IQA+ / MUSIQ on RealLQ250), **adding the SigLIP alignment alone** improves these scores to **0.600 / 0.620 / 62.000**, before any TLCM or large-scale data. Thus the SigLIP module is not a cosmetic plug-in, but an empirically important component in our caption-free setting.
> >
> > - **System-level benefits over caption-based UIR.**
> >
> >     Unlike prior UIR systems that rely on MLLM-generated captions, LucidFlux avoids both caption latency and caption bias. Table 3 shows that we remove the caption preprocessing stage entirely, which helps keep our end-to-end runtime competitive despite using a 12B backbone. In Appendix A.1 (Table 5), we further quantify that **17% and 24% of captions** produced by two representative MLLMs on RealLQ250 explicitly mention degradations, confirming that caption-based pipelines are systematically exposed to degradation-related textual bias. Our SigLIP-based semantic module, once projected through the Connector, sidesteps this issue by providing direct, caption-free semantics to the Flux.1 text interface.
> >
> >
> > Taken together, the SigLIP-based module is not just “another encoder”: it is the key enabler of a **practically deployable, caption-free semantic alignment mechanism** for large-DiT UIR, with both quantitative gains (Table 4) and clear system-level advantages (Table 3, Appendix A.1) over caption-dependent designs.

---

> ### Comment · Reviewer_aM2Z · 2025-11-27
> **Official Comment**
>
> Thanks for the detailed response. My questions have been addressed. I have thus raised my score to 6.

---

> > ### Author Response · Authors · 2025-11-27
> > **Thank you for your response**
> >
> > Thank you very much for your positive assessment of our work; your comments and suggestions have been helpful in improving its quality.

---

> ### Author Response · Authors · 2025-11-28
> **Thank you for your response**
>
> Dear Reviewer **aM2Z**,
>
> We are glad that our detailed response has addressed your questions and that you already raised the score to 6. We truly appreciate it.
>
> Best Regards,
>
> Authors of Paper #6814

---

### Author Response · Authors · 2025-11-28
**Summary of Rebuttal Consensus: Scores Improved to positive**

Dear Area Chair,

Given the recent decision to revert scores and assign new ACs, we provide this summary to clarify the **actual consensus** reached during the discussion period. Before the system freeze/reversion, the reviewers' assessment had evolved from an initial {8, 6, 4, 4} to a consensus of **Accept {8, 6, 6, Positive}**.

We respectfully request you consider the **explicit statements from reviewers** documented below, which reflect their updated views after our rebuttal:

**1. Reviewer aM2Z (Score: 4 ⇒ 6, Confirmed)**

- **Status:** The reviewer officially raised their score to **6** on Nov 28.
- **Reviewer's Statement:** *"Thanks for the detailed response. My questions have been addressed. I have thus raised my score to 6."* (Posted: 28 Nov 2025, 00:12 UTC)
- **Key Resolution:** We clarified the differences between our Dual-Branch Conditioner and DiffBIR, and added the requested comparisons.

**2. Reviewer 8VrS (Score: 4 ⇒ Intended Rise, Blocked by System)**

- **Status:** The reviewer explicitly stated their intention to raise the score but was technically blocked by the system freeze.
- **Reviewer's Statement:** *"Thank you for your response. It has addressed most of my concerns, and I am willing to raise my score. However, it seems that I am unable to edit my review and score at this moment."* (Posted: 28 Nov 2025, 12:04 UTC)
- **Key Resolution:** We addressed concerns regarding "Universal" terminology (revised title), architectural novelty (vs. DreamClear), and provided the requested inference efficiency comparisons.

**3. Reviewer V2EC (Score: 8)**

- **Status:** Maintained a strong **8**.
- **Assessment:** Praised the paper as *"well written"* with *"impressive qualitative/visual results"* and noted that *"LucidFlux seems to perform well compared to recent open- and closed-source methods."*

**4. Reviewer arXC (Score: 6, Marginally Above Acceptance)**

- **Status:** Maintained a positive **6**.
- **Assessment:** Acknowledged the contributions of the caption-free semantic alignment and the scalable data curation pipeline. We further provided the requested ablation studies (Caption-free vs. VLM/GT captions) which solidified their assessment.

**Summary**
The discussion was fruitful. **All four reviewers are now positively inclined towards the paper**. We hope this summary assists you in making an informed decision based on the final state of the review process.

Best regards,

Authors of Paper #6814

---

### Meta-Review · Area_Chair_Jfi2 · 2026-01-02

**Summary:**

This paper proposes an effective caption-free image restoration framework based on a large-scale diffusion transformer. It is interesting that the proposed method is caption-free. However, the main concerns of this paper include the limited contributions, inaccurate expressions, missing comparisons on the same Flux.1 backbone, and lacks of inference efficiency comparisons.

**Reviewer Concerns:**

The major concerns including the limited contributions, inaccurate expressions, missing comparisons on the same Flux.1 backbone, and lacks of inference efficiency comparisons should be addressed. More importantly, the novelty of the proposed method is minor compared with DreamClear and DiffBIR.

**Reviewer Scores:**

In the provided rebuttal, the authors add comparisons with DiffBIR method. The major differences from DiffBIR and DreamClear should be better clarified.

---

### Decision · Program_Chairs · 2026-01-26

Accept (Poster)